# Analytical Restructuring of Feed-Forward Networks for Accelerated LLM Inference

## Abstract

Scaling large language models (LLMs) improves performance but dramatically increases inference costs, with feed-forward networks (FFNs) consuming the majority of computational resources. While sparse architectures like mixture-of-experts (MoE) can mitigate this, inducing sparsity in existing dense models typically requires extensive, resource-intensive retraining (often hundreds of billions of tokens), creating a prohibitive barrier to practical deployment. We propose a broadly applicable post-training framework that improves this performance–cost trade-off by enabling the rapid, analytical restructuring of FFNs into a sparse, efficient architecture. The framework operates by analyzing neuron activation patterns from a small calibration dataset, then analytically rebuilding the FFN into a Mixture-of-Experts-style architecture with always-active "shared" experts and conditionally activated "routed" experts. Critically, this process can restructure dense FFNs into sparse MoE architectures and can also be applied recursively to the experts within existing MoE models to create finer-grained hierarchical sparsity for further acceleration. We construct a differentiable router directly from activation statistics, enabling immediate deployment with a useful training-free baseline and serving as a robust foundation for optional, lightweight fine-tuning. Experiments validate our approach across diverse settings, delivering practical speedups reaching up to $1.17\times$ in compute-bound scenarios while providing consistent gains across all configurations. This is achieved with only minutes of processing time and minimal fine-tuning (2k samples), which favorably contrasts with methods requiring orders of magnitude more computational resources. By providing an efficient, analytical path to high-performance sparsity, the framework makes accelerated LLM deployment practical and accessible for resource-constrained environments.

## 1 Introduction

Large language models (LLMs) have achieved remarkable performance on a wide range of tasks Zhang et al. (2022); Touvron et al. (2023); Liu et al. (2024b;a), but their ever-growing size presents significant deployment challenges due to high computational demands, especially on resource-constrained hardware or under strict latency budgets. This has spurred the development of various inference acceleration techniques. Methods like pruning Lu et al. (2024) and quantization Lin et al. (2024); Pei et al. (2023) are widely used but typically induce static changes to the model's architecture or numerical precision. A different paradigm, the mixture-of-experts (MoE) architecture Lepikhin et al. (2020); Du et al. (2022); Fedus et al. (2022); Dai et al. (2024), decouples model capacity from computational cost by using a router to dynamically select a sparse subset of parameters for each input token. However, reaping the benefits of MoE models has traditionally required expensive pre-training from scratch, establishing a challenging trade-off between model performance and training cost.

The computational bottleneck in modern transformer architectures is disproportionately located in the feed-forward network (FFN) blocks. Several studies have reported high activation sparsity in FFN neurons Liu et al. (2023); Zhang et al. (2021); Pei et al. (2024), meaning only a small fraction of neurons activate for any given input. This natural sparsity presents a compelling opportunity to accelerate inference without the cost of pre-training. A key research question is thus how to impose an efficient, structured sparsity pattern onto an already-trained model as a lightweight, post hoc optimization. While some prior work has explored restructuring dense models into MoEs, these methods perpetuate the costly training paradigm Zhu et al. (2024); Qu et al. (2024); Zheng

et al. (2024). They often require substantial, resource-intensive continual training (on the order of hundreds of billions of tokens) to recover model quality, creating a significant barrier to practical deployment. This high cost frames a critical gap: the need for a method that delivers the benefits of sparsity without the prohibitive computational expense.

To overcome these limitations, we propose an analytical post-training framework that improves the performance-cost trade-off for LLM acceleration. The framework restructures FFNs through a rapid, analytical process using only a tiny calibration dataset. It operates by analyzing neuron activation patterns to distinguish frequently active neurons (grouped into 'shared' experts) from sparsely active ones. The sparsely active neurons are then clustered into specialized 'routed' experts using a balanced assignment algorithm Jonker & Volgenant (1988). This restructuring is broadly applicable: it can transform a dense model's single, large FFN into a sparse MoE architecture, or it can be applied recursively to the individual experts of an existing MoE model to induce a finer-grained hierarchical sparsity. Crucially, the framework constructs a differentiable router analytically from activation statistics, bypassing the need for expensive router training and enabling rapid deployment with a strong training-free baseline or optional, lightweight fine-tuning.

Our contributions are:

- **A New Efficiency Paradigm for Sparsity:** We challenge the costly training-based approach by introducing an analytical, post hoc FFN restructuring method. Our framework achieves a superior trade-off between performance and conversion cost, making sparsity practical for rapid deployment.
- **Universal and Hierarchical Restructuring:** The proposed method rapidly restructures any FFN into a sparse, expert-based architecture. We demonstrate its universality by showing it can not only restructure dense models but also optimize existing MoE models by creating a hierarchical expert structure.
- **Analytical, Differentiable Router:** The router is initialized directly from activation statistics, providing immediate functionality and a strong starting point for optional fine-tuning, a key advantage over methods requiring router training from scratch.
- **Strong Performance with Practical Speedups:** The method achieves strong performance while delivering practical speedups reaching up to $1.17\times$ in compute-bound scenarios, requiring only minimal fine-tuning with 2k samples to rival methods that use orders of magnitude more compute.

The combination of speed, flexibility, and analytical construction presents a compelling and practical solution for researchers and practitioners seeking to deploy any LLM architecture more efficiently.

## 2 RELATED WORK

In contrast to pretraining MoE models from scratch, recent research has investigated the feasibility of constructing MoE architectures by repurposing existing dense LLMs. Current methodologies for deriving MoE models from dense checkpoints generally follow two paradigms: (1) partitioning parameters of FFNs while preserving the original model's total parameter count Zuo et al. (2022); Zhang et al. (2021); Yang et al. (2024), or (2) expanding the model's overall capacity while retaining activation dimensions comparable to standard dense models Komatsuzaki et al. (2022); Wu et al. (2024). This work prioritizes the former approach. Notably, MoEBERT Zuo et al. (2022) introduces an importance-driven strategy to transform FFNs into expert modules by strategically redistributing top-scoring neurons across specialized components. Concurrently, MoEfication Zhang et al. (2021) leverages the discovery of sparse activation patterns in ReLU-based FFNs within T5 architectures, enabling the decomposition of these layers into distinct expert groups governed by a learned routing mechanism. Based on continual training,Zhu et al. (2024) modifies the LLaMA-2 7B model as a LLaMA-MoE-3.5B MoE model, where the parameters of the original FFNs are partitioned into multiple experts. In Qu et al. (2024), based on a two-stage post-training strategy, an MoE model is constructed from the LLaMA3 8B model, where both attention and MLP are partitioned into MoE blocks. EMoE Qiu et al. (2023) creates MoE structures during fine-tuning by clustering neurons based on their key vectors, enabling conditional computation without adding extra parameters. Read-ME Cai et al. (2024) further explores refactorizing dense LLMs into MoE architectures with a decoupled router and system co-design, focusing on domain-aware expert construction and optimized inference (e.g., batching and caching). Compared to these approaches, our

method analytically restructures FFNs into experts by splitting neurons into shared and routed experts based on binary activation features and a balanced assignment objective, then constructing a router directly from representative neuron statistics. Under matched sparsity and a small 2k-sample fine-tuning budget, our analytical MoE restructuring achieves substantially better performance than random-initialized MLP baselines, while avoiding the heavy continual pretraining required by Zhu et al. (2024); Qu et al. (2024).

A parallel line of work studies fully differentiable routing and MoE training from scratch. Re-MoE Wang et al. (2024) replaces hard Top-K with ReLU routing and explicit load-balancing regularization, and Lory Zhong et al. (2024) performs segment-level differentiable expert merging trained on large token budgets. These methods focus on learning routers and experts jointly during pre-training, whereas our contribution is a training-light analytical restructuring of existing dense (or MoE) FFNs using only a tiny 2k-sample budget, making our approach complementary to differentiable-routing MoEs.

Orthogonal to FFN-to-MoE conversion, training-free activation sparsity methods such as TEAL Liu et al. (2024c) and WINA Chen et al. (2025) keep the backbone dense but sparsify neuron or channel activations at inference time using magnitude- or weight-informed thresholds plus specialized sparse kernels. These approaches operate at the neuron level and can in principle be applied inside MoE experts, whereas our method changes the architectural granularity by restructuring FFNs into shared and routed experts with an analytical router; this makes training-free activation sparsity complementary rather than competing with our Dense-to-MoE conversion.

## 3 METHODOLOGY

The proposed framework transforms dense LLMs into sparsely activated MoE architectures through two key phases: efficient expert grouping and analytical router construction. As shown in Figure 1, the framework operates through the following systematic process:

**A. Neuron Activation Profiling** (Section 3.1) Using a small calibration dataset, the framework profiles the activation patterns of neurons within each FFN layer to categorize them into shared experts (high-activation, task-agnostic) and routed experts (sparsely activated, task-specific).

**B. Expert Grouping** (Section 3.1) Shared Experts: Neurons exhibiting the highest activation rates are directly assigned to shared experts, which remain consistently activated during inference. Routed Experts: The remaining neurons are efficiently partitioned into routed experts through balanced clustering, mathematically formulated as a linear assignment problem.

**C. Router Construction and Optimization** (Section 3.2) The routing mechanism is analytically derived from the activation statistics of representative neurons in each expert cluster, with differentiable enhancements and load balancing for fine-tuning scenarios.

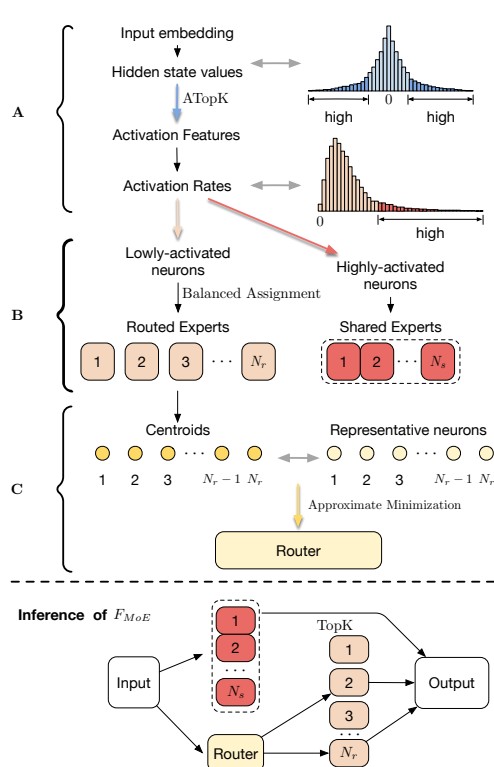

Figure 1: Overview of the proposed analytical FFN-to-MoE restructuring framework.

### 3.1 SHARED AND ROUTED EXPERTS GROUPING

An FFN layer computes $F(\mathbf{x})$ and adds it to the input embedding $\mathbf{x}$ via a residual connection. For LLaMA models with SwiGLU, this function is:

$$F(\mathbf{x}) = \mathbf{W}_{\text{down}}^{\top}\mathbf{h}, \quad \text{where } \mathbf{h} = \text{Swish}(\mathbf{W}_{\text{gate}}^{\top}\mathbf{x}) \odot (\mathbf{W}_{\text{up}}^{\top}\mathbf{x}). \tag{1}$$

Here, $\mathbf{x} \in \mathbb{R}^d$, $\mathbf{W}_{\text{up}}, \mathbf{W}_{\text{gate}} \in \mathbb{R}^{d \times d_h}$, and $\mathbf{W}_{\text{down}} \in \mathbb{R}^{d_h \times d}$, where $d_h$ is the FFN hidden width. Our goal is to restructure this dense FFN into a sparse expert-based architecture composed of $N_s$ shared experts and $N_r$ routed experts ($N_s + N_r = N$). The final output is a combination of a single shared expert $E^s$ and the Top-K selected routed experts $E_i^r$:

$$F_{MoE}(\mathbf{x}) = E^s(\mathbf{x}) + \sum_{i=1}^{N_r} g_i(\mathbf{x}) \cdot E_i^r(\mathbf{x}), \tag{2}$$

where $g_i(\mathbf{x})$ is the gate value from a router network $G$. During inference, only the top $N_k$ routed experts (by router score) are activated and others receive gate value 0, where $N_k$ is the number of routed experts activated per token. The following sections detail how we construct these experts from the original FFN weights without training.

### 3.1.1 SHARED EXPERTS: IDENTIFYING GLOBAL PATTERNS

Shared experts are designed to capture common knowledge by housing neurons that are consistently active. To identify them, we first analyze neuron activations on a small calibration dataset. We compute the hidden states $\mathbf{H} \in \mathbb{R}^{q \times d_h}$ for a batch of $q$ tokens.

Due to FFN activation sparsity, a neuron's importance can be measured by its activation frequency. For each token, we identify the $K_a$ neurons with the highest activation magnitudes. This yields a binary activation matrix $\mathbf{A} = [\mathbf{c}_1 \ \mathbf{c}_2 \ \cdots \ \mathbf{c}_{d_h}] \in \{0, 1\}^{q \times d_h}$, where each column $\mathbf{c}_i$ is the **activation feature vector** for neuron $i$, representing its firing pattern across the dataset. A neuron's activation rate, $\mu_i$, is the mean of its feature vector $\mathbf{c}_i$. The sparsity analysis is detailed in Appendix A.1, while the complete step-by-step pipeline is provided in Appendix A.2. (Here $K_a$ is a small calibration hyperparameter.)

We select the $N_s \cdot m$ neurons with the highest activation rates ($\mu_i$) to form the shared expert $E^s$. (Here $m$ denotes the per-expert neuron count.) Its weights ($\mathbf{W}_{\text{up}}^s, \mathbf{W}_{\text{gate}}^s, \mathbf{W}_{\text{down}}^s$) are constructed by slicing the original FFN weight matrices according to the selected neuron indices.

### 3.1.2 ROUTED EXPERTS: CLUSTERING SPECIALIZED PATTERNS

Routed experts handle specialized, input-dependent computations. We form them by grouping the remaining neurons based on functional similarity. Our key insight is that neurons with similar roles will have similar activation patterns.

Therefore, we cluster the remaining neurons by applying a **balanced assignment algorithm** to their activation feature vectors ($\mathbf{c}_i$) from the previous step. This algorithm groups neurons into $N_r$ experts, each of size $m$, by minimizing intra-cluster distance based on their co-activation patterns. The detailed optimization is provided in Appendix A.3. Each routed expert $E_p^r$ is then constructed by slicing the original FFN weights with its assigned neuron indices.

### 3.2 ROUTER CONSTRUCTION AND OPTIMIZATION

To preserve knowledge from the dense FFN, we design a router $G$ that predicts the importance of each expert such that $F_{MoE}(\mathbf{x})$ approximates the original output $F(\mathbf{x})$. We conceptualize the dense FFN as an MoE where all experts are active: $F(\mathbf{x}) = \sum_{i=1}^{N} E_i(\mathbf{x})$.

Our approach identifies a single **representative neuron** $R_j$ to act as a proxy for each expert $j$. This neuron is the one whose activation feature vector $\mathbf{c}_{R_j}$ (from matrix $\mathbf{A}$) is closest to the expert's cluster centroid $\hat{\mathbf{c}}_j$. Since the centroid embodies the expert's average activation pattern, this neuron serves as an ideal proxy. The router is constructed using only the parameters of these representative neurons:

$$G(\mathbf{x}) = \text{Swish}(\mathbf{x}\mathbf{W}_{\text{gate}}^R) \odot (\mathbf{x}\mathbf{W}_{\text{up}}^R), \tag{3}$$

where $\mathbf{W}_{\text{gate}}^R = \mathbf{W}_{\text{gate}}[:, S_R]$, $\mathbf{W}_{\text{up}}^R = \mathbf{W}_{\text{up}}[:, S_R]$, and $S_R = \{R_1, \ldots, R_{N_r}\}$ contains the representative neuron indices. This yields router scores $\mathbf{s} = [s_1, \ldots, s_{N_r}]$ reflecting each expert's expected contribution. The detailed mathematical derivation is provided in Appendix A.4.

The initial router uses hard Top-K selection, which is non-differentiable. To enable optional fine-tuning, we introduce learnable scaling parameters $\mathbf{u} = [u_1, \ldots, u_{N_r}]$, initialized to zero. For a

selected expert $i$, the binary gate value of 1 is replaced by a soft gate $1 + s_i' \cdot u_i$, where $\mathbf{s}' = \text{Softmax}(\mathbf{s})$. This allows the model to learn to modulate expert contributions while preserving the initial performance. To ensure balanced expert utilization without auxiliary losses, we introduce adaptive bias terms $\mathbf{b} = [b_1, \dots, b_{N_r}]$ added to scores before Top-K selection. The final gating logic is:

$$g_i = \begin{cases} 1 + s_i' \cdot u_i, & s_i' + b_i \in \text{Top-K}(\{s_j' + b_j \mid 1 \le j \le N_r\}, N_k), \\ 0, & \text{otherwise} \end{cases} \qquad (4)$$

During fine-tuning, we update the adaptive biases $\mathbf{b}$ using a simple utilization-based controller. For each layer and step, let $T$ be the number of routed tokens and $N_k$ the number of routed experts activated per token; the load of expert $i$ is $L_i$ (number of times it appears in the Top-$N_k$ set), and its empirical utilization is $p_i = L_i / (N_k T)$. With a uniform utilization target $p^* = 1/N_r$ and a small step size $\gamma$ (we use $\gamma = 10^{-3}$), the bias update is $b_i \leftarrow b_i + \gamma(p^* - p_i)$. Overloaded experts ($p_i > p^*$) are gradually down-biased, while under-utilized experts ($p_i < p^*$) are up-biased, increasing utilization entropy and reducing load variance without introducing an auxiliary load-balancing loss. At inference, the router operates on the fixed learned $\mathbf{u}$ and $\mathbf{b}$ with negligible overhead because it only processes representative neurons.

## 3.3 Application to Existing MoE Models

The framework's analytical approach is broadly applicable, allowing it not only to restructure dense FFNs into MoEs but also to optimize existing MoE models by inducing a finer-grained, hierarchical sparsity. This is achieved by applying the restructuring process to each expert within an MoE layer individually.

Consider a standard MoE layer where the output is a gated sum of expert outputs: $F_{MoE}(\mathbf{x}) = \sum_{i=1}^{N_r} g_i(\mathbf{x}) \cdot E_i(\mathbf{x})$, where $g_i(\mathbf{x})$ is the gate value for expert $E_i$. Each expert $E_i$ is itself a standard FFN. Our intra-expert restructuring applies the methodology described in Section 3.1 and Section 3.2 to each of these expert FFNs.

This transforms each original expert $E_i$ into its own hierarchical expert structure, containing one always-active shared sub-expert ($E_i^s$) and a set of routable, specialized sub-experts ($E_{i,j}^r$). The output of the original expert $E_i$ is thus reformulated as:

$$E_i(\mathbf{x}) \rightarrow E_i'(\mathbf{x}) = E_i^s(\mathbf{x}) + \sum_{j=1}^{N_r'} g_{i,j}'(\mathbf{x}) \cdot E_{i,j}^r(\mathbf{x}), \qquad (5)$$

where $g_{i,j}'(\mathbf{x})$ are the gate values from a newly constructed sub-router for the sub-experts within $E_i$.

The final output of the entire MoE layer becomes a two-level hierarchy: a top-level router selects which primary experts to activate, and within each activated expert, a second-level sub-router selects which specialized sub-experts to use. This induces a more profound and dynamic sparsity, further accelerating inference by ensuring that only a small fraction of neurons within the already-selected experts are utilized. This application underscores the approach as a general-purpose FFN restructuring framework for maximizing computational efficiency.

## 4 Experiments

We evaluate the proposed framework as a post-training sparsification method for inference acceleration on large language models. Our implementation uses Hugging Face Transformers Wolf (2019) and PyTorch Paszke et al. (2019).

### 4.1 Main Results

**Calibration:** We randomly select 8 examples (2048 sequence length) from WikiText-2 Merity et al. (2016) to compute activation statistics for neuron grouping and initial router construction. We set $K_a = 10$ for the activation status record.

**Lightweight Fine-tuning (2k):** We fine-tune using LoRA Hu et al. (2021) (rank 8, alpha 32) on 2,048 WikiText-2 samples for 1 epoch. The optimizer is Adam Kingma (2014) ($\beta_1 = 0.9, \beta_2 =$

0.95). We use different learning rates for the router scale parameter (0.001) and other LoRA parameters (5.95e-5). The load balancing bias update speed $\gamma = 0.001$.

We compare our method against several approaches for accelerating LLM inference: (1) **Dense Models:** Original Llama-2 7B, Llama-2 70B, Qwen-2.5-7B, and Qwen-3-30B-A3B checkpoints serve as performance upper bounds. (2) **Structured Pruning:** SliceGPT Ashkboos et al. (2024) and SLEB Song et al. (2024), which remove structured components (20% reduction). We use 20% pruning for fair comparison since these methods prune the entire model structure while our method only sparsifies FFN layers. (3) **MoE-restructuring:** LLaMA-MoE Zhu et al. (2024), LLaMA-MoE-v2 Qu et al. (2024), and EMoE Qiu et al. (2023), which restructure dense FFNs into sparse MoE architectures.

The main results use 25% sparsity with S3A3E8 configuration (3 shared + 3 active routed / 8 total), balancing performance and efficiency. For fair comparison, we configure all MoE methods with 8 total experts by default.

Table 1: Downstream task accuracy (zero-shot evaluation) after LoRA fine-tuning on 2k WikiText-2 samples. Higher is better. We use 25% sparsity with a 1:1 shared/routed expert configuration.

| Method | Sparsity | Type | PIQA | WinoGrande | ARC-E | ARC-C | HellaSwag |
|---|---|---|---|---|---|---|---|
| **Llama-2 7B** | | | | | | | |
| Dense | 0% | - | 78.78 | 69.06 | 74.58 | 46.16 | 76.00 |
| SliceGPT | 20% | Structured Pruning | 65.71 | 62.88 | 59.76 | 33.21 | 51.34 |
| SLEB | 20% | Structured Pruning | 73.13 | 58.98 | 57.90 | 33.02 | 62.47 |
| LLaMA-MoE | 25% | MoE Restructuring | 49.35 | 50.28 | 54.04 | 26.37 | 25.77 |
| LLaMA-MoE-v2 | 25% | MoE Restructuring | 63.55 | 59.35 | 63.77 | 34.81 | 54.89 |
| EMoE | 25% | MoE Restructuring | 72.47 | 64.48 | 58.63 | 35.75 | 60.80 |
| **Ours** | 25% | MoE Restructuring | **74.34** | **65.77** | **67.09** | **40.35** | **69.36** |
| **Llama-2 70B** | | | | | | | |
| Dense | 0% | - | 82.70 | 77.98 | 80.98 | 57.34 | 83.84 |
| SliceGPT | 20% | Structured Pruning | 68.91 | 70.06 | 64.56 | 41.14 | 56.26 |
| SLEB | 20% | Structured Pruning | 77.39 | 65.55 | 62.37 | 40.11 | 68.39 |
| LLaMA-MoE | 25% | MoE Restructuring | 51.95 | 56.50 | 59.09 | 32.40 | 27.57 |
| LLaMA-MoE-v2 | 25% | MoE Restructuring | 66.79 | 66.57 | 68.94 | 42.38 | 59.57 |
| EMoE | 25% | MoE Restructuring | 76.34 | 72.33 | 63.47 | 43.62 | 66.19 |
| **Ours** | 25% | MoE Restructuring | **78.49** | **73.49** | **73.32** | **49.86** | **76.12** |
| **Qwen-2.5-7B** | | | | | | | |
| Dense | 0% | - | 79.82 | 73.16 | 77.36 | 51.02 | 78.86 |
| SliceGPT | 20% | Structured Pruning | 66.19 | 66.51 | 61.88 | 36.69 | 53.21 |
| SLEB | 20% | Structured Pruning | 74.95 | 61.76 | 59.95 | 35.80 | 64.41 |
| LLaMA-MoE | 25% | MoE Restructuring | 49.63 | 53.21 | 57.05 | 28.64 | 25.65 |
| LLaMA-MoE-v2 | 25% | MoE Restructuring | 64.25 | 62.71 | 65.77 | 37.59 | 56.06 |
| EMoE | 25% | MoE Restructuring | 73.98 | 65.41 | 60.63 | 38.48 | 62.71 |
| **Ours** | 25% | MoE Restructuring | **75.93** | **69.36** | **70.59** | **43.86** | **72.21** |
| **Qwen-3-30B-A3B** | | | | | | | |
| Dense | 0% | - | 84.51 | 79.18 | 84.43 | 57.88 | 87.44 |
| SliceGPT | 20% | Structured Pruning | 70.60 | 71.58 | 66.88 | 41.85 | 58.41 |
| SLEB | 20% | Structured Pruning | 79.16 | 66.01 | 70.08 | 42.11 | 71.74 |
| LLaMA-MoE | 25% | MoE Restructuring | 52.18 | 54.48 | 62.50 | 30.77 | 28.32 |
| LLaMA-MoE-v2 | 25% | MoE Restructuring | 65.54 | 67.24 | 71.27 | 41.99 | 62.78 |
| EMoE | 25% | MoE Restructuring | 74.76 | 70.50 | 65.78 | 43.12 | 70.62 |
| **Ours** | 25% | MoE Restructuring | **80.23** | **74.84** | **76.75** | **48.80** | **80.71** |

**Evaluation on Downstream Task Performance**. Table 1 presents zero-shot results on five common benchmarks: PIQA Bisk et al. (2020), WinoGrande Sakaguchi et al. (2021), ARC-Easy, ARC-Challenge Clark et al. (2018), and HellaSwag Zellers et al. (2019). At 25% sparsity, the proposed method consistently outperforms all baseline methods across four different base models. On Llama-2 7B, it achieves 74.34% on PIQA and 69.36% on HellaSwag, substantially exceeding both structured pruning methods and other MoE restructuring approaches. The effectiveness generalizes across model scales and architectures: on the larger Qwen-3-30B-A3B model, it achieves

Table 2: Broader downstream evaluation on Llama-2 7B at 25% sparsity (S3A3E8). We report MMLU-5shot, HumanEval pass@1, and GSM8K-8shot (higher is better).

| Method | MMLU-5shot (%) | HumanEval pass@1 (%) | GSM8K-8shot (%) |
|---|---|---|---|
| Dense | 45.81 | 12.72 | 14.31 |
| LLaMA-MoE | 35.09 | 7.58 | 7.41 |
| LLaMA-MoE-v2 | 38.02 | 9.32 | 10.09 |
| EMoE | 43.11 | 10.29 | 12.55 |
| **Ours** | **44.02** | **11.22** | **13.01** |

Table 3: Matched-budget clustering and routing comparison on Llama-2 7B (MMLU-5shot, higher is better). All methods use 25% sparsity, identical expert counts, and 2k-sample fine-tuning.

| Method | Expert grouping | Router | MMLU-5shot (%) | $\Delta$ vs Dense (pp) |
|---|---|---|---|---|
| Dense | - | - | 45.81 | +0.00 |
| MoEfication (budget-matched) | Parameter K-means | MLP router | 35.17 | -10.64 |
| READ-ME (budget-matched) | Domain-aware clustering | Global router | 31.24 | -14.57 |
| MoEfication-clustering + ours | Parameter K-means | Analytical router | 37.33 | -8.48 |
| READ-ME-clustering + ours | Domain-aware clustering | Analytical router | 36.79 | -9.02 |
| **Ours (analytical)** | Binary-activation balanced assign. | Analytical router | **44.02** | **-1.79** |

80.23% on PIQA and 80.71% on HellaSwag, demonstrating robust performance improvements even on state-of-the-art foundation models. The results demonstrate that the analytical construction and lightweight fine-tuning enable effective sparsification while maintaining competitive performance across diverse model architectures and scales.

**Broader Evaluation on Knowledge, Coding, and Math**. Beyond these five zero-shot tasks, we also evaluate Llama-2 7B at 25% sparsity (S3A3E8) on MMLU-5shot, HumanEval pass@1, and GSM8K-8shot to cover knowledge-intensive and reasoning benchmarks. As summarized in Table 2, our analytical MoE restructuring achieves 44.02% MMLU-5shot (only 1.79 pp below dense) and competitive coding/math accuracy, while MoEfication-style and LLaMA-MoE variants incur substantially larger drops, underscoring the robustness of our conversion on harder tasks.

**Matched-Budget Comparison with MoEfication and Read-ME**. Table 3 reports MMLU-5shot on Llama-2 7B under 25% sparsity with identical expert counts and a 2k-sample fine-tuning budget for all MoE conversions. Our analytical MoE restructuring reaches 44.02% MMLU-5shot (only 1.79 pp below the dense baseline at 45.81%), whereas budget-matched MoEfication and READ-ME variants remain 8–15 pp below dense. Using the same router, switching from K-means or domain-aware clustering to our binary-activation balanced clustering yields an additional +6.69 pp, highlighting the importance of the shared–routed split and balanced assignment.

## 4.2 ABLATION STUDIES

**Efficient Fine-tuning: Achieving Strong Performance with Minimal Data**. Figure 2 demonstrates the method's capability for rapid deployment and data-efficient adaptation with the 25% sparsity configuration. The approach achieves strong performance immediately after construction with zero fine-tuning data, showcasing the effectiveness of the analytical router initialization from activation statistics. This training-free performance provides practical value, enabling quick deployment without adaptation overhead. Building upon this solid foundation, the method achieves further substantial performance recovery with as few as 1,024 WikiText-2 samples, reaching near-optimal results that plateau quickly with additional data. This rapid convergence from an already strong baseline showcases the effectiveness of the analytical construction: the method requires minimal fine-tuning because the initial router initialization already captures essential activation patterns. This analysis highlights practical advantages: delivering competitive sparsification directly after construction and achieving strong performance with minimal computational overhead, making it suitable for industrial deployment where extensive retraining is prohibitive.

To further isolate the role of lightweight fine-tuning, Table 4 compares our method with LLaMA-MoE-v2 on Llama-2 7B under the same 25% sparsity. Our training-free model already achieves 42.50% MMLU-5shot with reasonable perplexity (7.32/11.98 on Wiki/C4), outperforming LLaMA-MoE-v2 even after fine-tuning (34.81%, 8.68/19.76). With only 2k samples, our fine-tuned model

Table 4: Training-free vs fine-tuned comparison on Llama-2 7B and LLaMA-MoE-v2 (25% sparsity; identical decoding). We report MMLU-5shot (higher is better) and language modeling perplexity (PPL; lower is better).

| Model | Regime | MMLU-5shot (%) | PPL-Wiki | PPL-C4 |
|---|---|---|---|---|
| LLaMA-MoE-v2 | Training-free | 30.33 | > 10,000 | > 7,000 |
| LLaMA-MoE-v2 | Fine-tuning | 34.81 | 8.68 | 19.76 |
| Ours | Training-free | 42.50 | 7.32 | 11.98 |
| Ours | Fine-tuning (2k) | **44.02** | **5.92** | **11.21** |

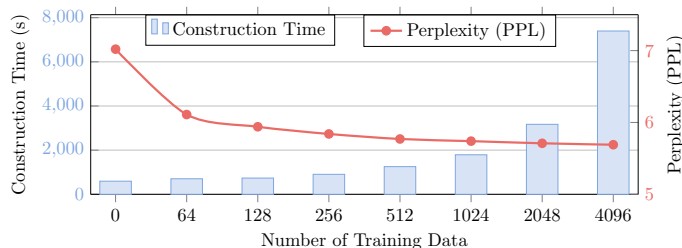

Figure 2: Data efficiency: Model performance and construction time with increasing fine-tuning data (WikiText-2 samples, 25% sparsity).

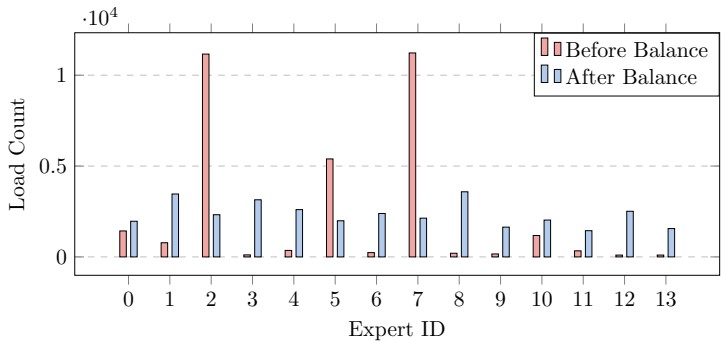

Figure 3: Load balancing effectiveness: Achieving uniform expert utilization.

reaches 44.02% MMLU-5shot and further reduces perplexity to 5.92/11.21, indicating that most of the gain comes from analytical restructuring, with fine-tuning acting as a small refinement.

**Effective Load Balancing for Better Expert Utilization**. Figure 3 demonstrates a sophisticated load balancing mechanism, which addresses a critical challenge in MoE architectures: expert utilization imbalance. Without load balancing, the final layer of Llama-2 7B exhibits severe activation skew, with some experts receiving disproportionately high traffic while others remain underutilized. The analytical load balancing technique effectively redistributes computational load across all experts, maximizing hardware efficiency and preventing bottlenecks. This balanced utilization is crucial for achieving the full speedup potential in industrial deployment, as imbalanced expert usage can lead to memory inefficiencies and reduced throughput. The uniform expert distribution showcased in the figure directly translates to more predictable and consistent inference performance.

**Efficiency: Token Budget and Conversion Time**. Table 5 summarizes the supervised token budgets and conversion times of our method and LLaMA-MoE variants. While LLaMA-MoE-v1 and v2 require 200B and ∼7B supervised tokens respectively (months- and days-scale continual training), our analytical restructuring uses only 2k samples and completes in 2,741s end-to-end (271s for the analytical construction itself), validating the "minutes-level" conversion claim.

**Calibration Sensitivity and Harder Benchmarks**. Table 6 examines MMLU-5shot and perplexity on Llama-2 7B under 25% sparsity while varying calibration source (WikiText-2 vs C4) and calibration set size. Increasing the calibration set from 8 to 64 samples yields modest MMLU gains

Table 5: Supervised token budget and conversion time for constructing MoE models. We report the supervised data required to obtain a usable MoE model, and (for ours) the measured end-to-end and analytical construction time on our setup.

| Method | Supervised token budget | End-to-end time | Construction time (ours) |
|---|---|---|---|
| **Ours** | **2k samples** | **2741s** | **271s** |
| LLaMA-MoE-v1 | 200B tokens | Months | 334s[†] |
| LLaMA-MoE-v2 | ∼7B tokens | Days | 509s[†] |

[†] Split-only time measured on our setup; reported training tokens from the original papers are not included.

Table 6: Calibration sensitivity on Llama-2 7B at 25% sparsity (S3A3E8). We vary calibration source (WikiText-2 vs C4) and calibration set size $n$ (number of samples). We report MMLU-5shot (higher is better), absolute drop vs dense ($\Delta$ pp; dense = 45.81), and perplexity (PPL; lower is better) on Wiki and C4.

| Calibration source | $n$ (samples) | MMLU-5shot (%) | $\Delta$ vs Dense (pp) | PPL-Wiki / PPL-C4 |
|---|---|---|---|---|
| WikiText-2 | 8 | 44.02 | -1.79 | 5.92 / 11.21 |
| WikiText-2 | 32 | 44.63 | -1.18 | 5.72 / 11.15 |
| WikiText-2 | 64 | **44.89** | **-0.92** | **5.69** / 10.98 |
| C4 | 8 | 42.31 | -3.50 | 7.04 / 9.17 |
| C4 | 32 | 43.25 | -2.56 | 6.92 / 9.07 |
| C4 | 64 | **43.39** | **-2.42** | 6.78 / **9.02** |

Table 7: Hierarchical application to an existing MoE model (Qwen3-30B-A3B) at 25% sparsity. We report GFLOPs per decoding step, GMACs per token, tokens per second, and MMLU-5shot.

| Method | GFLOPs ($\downarrow$) | GMACs ($\downarrow$) | tokens/s ($\uparrow$) | MMLU-5shot (%) |
|---|---|---|---|---|
| Dense | 778.7 | 389.33 | 1.19 | 80.78 |
| Ours (hierarchical) | 634.9 | 331.32 | 1.36 | 78.21 |

(e.g., 44.02→44.89 on WikiText-2) and small perplexity reductions; C4 calibration behaves similarly and sometimes slightly better on C4 PPL. Overall, these results indicate that our analytical pipeline is robust to calibration choices and achieves competitive MMLU with tiny calibration and tuning budgets.

**Hierarchical Application to Existing MoE Layers**. To empirically validate the hierarchical MoE application, we apply our intra-expert restructuring to an existing MoE model, Qwen3-30B-A3B, by splitting each expert into 8 sub-experts (width-splits) and reusing the S3A3E8 configuration inside each expert. As shown in Table 7, the resulting two-level hierarchy reduces GFLOPs by 18.5% and GMACs by 14.9%, while increasing throughput by 14.3% with only a 2.57 pp drop in MMLU-5shot. This demonstrates that our analytical construction extends beyond dense-to-MoE conversion and can induce beneficial hierarchical sparsity in existing MoE layers.

## 5 CONCLUSION

We introduced a post-training framework that improves the trade-off between performance and computational cost in deploying sparse LLMs. By analytically restructuring FFNs based on neuron activation statistics, the method efficiently remodels dense networks into high-performing sparse MoE architectures. This process requires only a tiny calibration dataset and minutes of computation, directly challenging the expensive, training-heavy paradigm of prior methods. Our key innovation lies in the analytical construction of both the expert partitions and the router, which enables strong performance out of the box and serves as a robust starting point for optional, lightweight fine-tuning. Furthermore, we demonstrated broad applicability by showing it can be applied not only to dense models but also to existing MoE models, creating a finer-grained hierarchical sparsity for further acceleration. This work makes performant, sparse LLMs more accessible and practical for a wide range of real-world applications.

## DECLARATION OF LLM USAGE

The usage of LLMs is strictly limited to aid and polish the paper writing.

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

## A   DETAILED MATHEMATICAL DERIVATIONS

This appendix provides the detailed mathematical derivations and algorithmic analysis that support the core concepts presented in the main manuscript.

### A.1   ACTIVATION SPARSITY ANALYSIS AND HYPOTHESIS

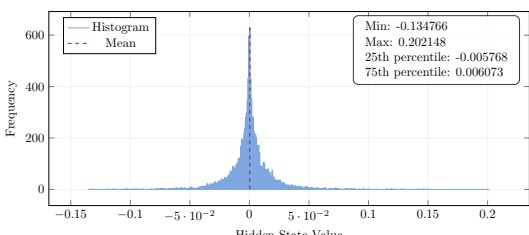

(a) The histogram of FFN hidden state $\mathbf{h}$ for the 3-th block and the $1,000$-th token.

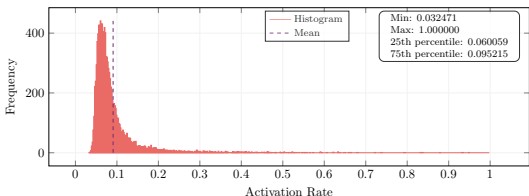

(b) The histogram of activation rates $\boldsymbol{\mu}$ for the 3-th block with $K_a = 1,000$.

Figure A.1: Empirical analysis of FFN activation patterns supporting our mathematical framework.

As demonstrated in fig. 1(a), the distribution of the FFN hidden state $\mathbf{h}$ is sharply peaked at $0$ and constrained within a small range. This indicates that most $h_i$ are concentrated near zero, confirming the sparsity of activations.

**Detailed Hypothesis and Derivation.** Given the input embedding $\mathbf{x} \in \mathbb{R}^d$, each neuron's contribution can be analyzed independently. For the $i$-th neuron:

$$h_i = \text{Swish}(\mathbf{x} \cdot \mathbf{w}_{gate,i}) \cdot (\mathbf{x} \cdot \mathbf{w}_{up,i})$$

where $\mathbf{w}_{gate,i}$ and $\mathbf{w}_{up,i}$ are the $i$-th columns of the gate and up projection weights. The FFN output decomposes as:

$$F(\mathbf{x}) = \sum_{i=1}^{d_h} h_i \mathbf{w}_{down,i} \tag{6}$$

Each $h_i$ acts as a gating score for the corresponding output weight $\mathbf{w}_{down,i}$. Since structured pruning research shows that $||F(\mathbf{x})||$ is typically small due to residual connections, we observe high sparsity in FFN activations. This leads to our central hypothesis:

$$\arg\min_i |h_i \mathbf{w}_{down,i}| \approx \arg\min_i |h_i| \tag{7}$$

This approximation is justified because when $h_i$ is extremely small, the product $h_i \mathbf{w}_{down,i}$ vanishes regardless of the magnitude of $\mathbf{w}_{down,i}$. The empirical evidence in Figure 1(a) supports this hypothesis by showing the high concentration of hidden states near zero.

### A.2   COMPLETE ACTIVATION ANALYSIS PIPELINE

To systematically quantify neuron activation patterns, we establish the complete mathematical pipeline starting from calibration data.

**Step 1: Tensor Reshaping and Hidden State Computation.** Given a batched input tensor $\mathbf{X} \in \mathbb{R}^{b \times s \times d}$ from the calibration dataset, where $b$ is batch size and $s$ is sequence length, we first reshape it to $\mathbf{X}' \in \mathbb{R}^{q \times d}$ where $q = b \cdot s$ is the total number of tokens. We then compute the hidden states:

$$\mathbf{H} = \text{Swish}(\mathbf{X}'\mathbf{W}_{gate}) \odot (\mathbf{X}'\mathbf{W}_{up}) \in \mathbb{R}^{q \times d_h} \tag{8}$$

Note that in practical implementation, we normalize $\mathbf{X}'$, $\mathbf{W}_{gate}$ and $\mathbf{W}_{up}$ before the calculation to eliminate the influence of their magnitudes on the output.

**Step 2: Activation Matrix Construction.** Using the ATopK metric from the main text, we apply it row-wise to the hidden state matrix $\mathbf{H}$ to create binary activation markers, directly producing the activation matrix $\mathbf{A} \in \mathbb{R}^{q \times d_h}$:

$$\mathbf{A} = [\mathbf{c}_1 \ \mathbf{c}_2 \ \cdots \ \mathbf{c}_{d_h}]$$

where each column $\mathbf{c}_i \in \mathbb{R}^q$ is the activation feature vector representing neuron $i$'s activation status across all $q$ calibration tokens, and $\mathbf{A}[t, i] = a_{t,i}$ as defined by the ATopK metric.

**Step 3: Activation Rate Computation.** The activation rates are computed by averaging each column of the activation matrix:

$$\boldsymbol{\mu} = [\mu_1, \mu_2, \cdots, \mu_{d_h}], \text{ where } \mu_i = \frac{1}{q} \sum_{t=1}^{q} \mathbf{A}[t, i] = \text{mean}(\mathbf{c}_i) \tag{9}$$

The histogram of these activation rates $\boldsymbol{\mu}$ is shown in Figure 1(b). The histogram reveals a highly skewed distribution of activation rates, where the majority of neurons exhibit low activation rates (below 0.1), with a sharp peak near 0.07. However, the distribution also features a long tail, indicating the presence of a subset of neurons with significantly higher activation rates extending up to 1. These high-activation neurons are likely active across a wide range of input tokens, making them suitable for processing common knowledge rather than task-specific specialization. Therefore, we identify neurons for shared experts by grouping these high-activation neurons. Given the total number of shared experts as $N_s$ and the expert size $m$, we get the selection indices set $S_{N_s}$ by selecting $N_s \cdot m$ neurons with highest activation rates based on $\boldsymbol{\mu}$:

$$S_{N_s} = \{i \ : \ \mu_i \in \text{TopK}(\{\mu_j \mid 1 \le j \le d_h\}, N_s \cdot m)\}. \tag{10}$$

These indices $S_{N_s}$ are then used to form the shared experts by assigning the corresponding parameters from the original FFN, as detailed in Section 3.1.

The majority of low activation rates also encourage us to construct routed experts, which are not always activated but are specialized for tokens encountered.

### A.3 DETAILED BALANCED CLUSTERING ALGORITHM FOR ROUTED EXPERTS

To construct routed experts, we employ a constrained balanced K-means clustering algorithm on the activation feature vectors $\mathbf{c}_i$ derived from matrix $\mathbf{A}$.

**Centroid Initialization.** We first identify $N_r$ centroids by selecting neurons (excluding those already assigned to shared experts) with the highest activation rates from $\boldsymbol{\mu}$:

$$C = \{\mathbf{c}_i : \mu_i \in \text{TopK}(\mu_j \mid 1 \le j \le d_h, j \notin S_{N_s}, N_r)\} = \{\hat{\mathbf{c}}_1, \ldots, \hat{\mathbf{c}}_{N_r}\}$$

**Distance Matrix Construction.** We formalize the clustering by constructing a distance matrix $\mathbf{D} \in \mathbb{R}^{N_r \cdot m \times N_r}$, where element $d_{i,j}$ represents the $L_2$ distance between the $i$-th activation feature vector $\mathbf{c}_i$ and the $j$-th centroid $\hat{\mathbf{c}}_j$:

$$d_{i,j} = \|\mathbf{c}_i - \hat{\mathbf{c}}_j\|_2 = \sqrt{\sum_{k=1}^{q} (c_{k,i} - \hat{c}_{k,j})^2} \tag{11}$$

The constrained balanced K-means algorithm proceeds iteratively with centroids $\hat{\mathbf{c}}_1^t, \hat{\mathbf{c}}_2^t, \ldots, \hat{\mathbf{c}}_{N_r}^t$ at iteration $t$:

**Cluster Assignment:** Let $T_{i,p}^t$ be a solution to the following linear program:

$$\min_T \sum_{i=1}^{mN_r} \sum_{p=1}^{N_r} T_{i,p} \cdot d_{i,p} \tag{12}$$

$$\text{s.t. } \sum_{i=1}^{mN_r} T_{i,p} = m, \forall p \in \{1, \ldots, N_r\}; \ \sum_{p=1}^{N_r} T_{i,p} = 1, \ \forall i \in \{1, \ldots, mN_r\}; \ T_{i,p} \geq 0, \forall p, i.$$

**Cluster Update:**

$$\hat{\mathbf{c}}_p^{t+1} = \begin{cases} \frac{\sum_{i=1}^{N_r \cdot m} T_{i,p}^t \cdot \mathbf{c}_i}{\sum_{i=1}^{N_r \cdot m} T_{i,p}^t}, & \text{if } \sum_{i=1}^{N_r \cdot m} T_{i,p}^t > 0, \\ \hat{\mathbf{c}}_p^t, & \text{otherwise.} \end{cases} \tag{13}$$

Since this is an unbalanced assignment problem ($mN_r > m$), we reduce it to a balanced assignment by extending the distance matrix:

$$\mathbf{D}^{ext} = \big[ \underbrace{\mathbf{d}_1, \ldots, \mathbf{d}_1}_{m \text{ times}}, \underbrace{\mathbf{d}_2, \ldots, \mathbf{d}_2}_{m \text{ times}}, \ldots, \underbrace{\mathbf{d}_{N_r}, \ldots, \mathbf{d}_{N_r}}_{m \text{ times}} \big]$$

The balanced assignment problem becomes:

$$\min_{T'} \sum_{i=1}^{mN_r} \sum_{p'=1}^{N_r \cdot m} T_{i,p'}' \cdot d_{i,p'}^{ext} \tag{14}$$

$$\text{s.t. } \sum_{i=1}^{mN_r} T_{i,p'}' = 1, \forall p' \in \{1, \ldots, mN_r\}; \ \sum_{p'=1}^{mN_r} T_{i,p'}' = 1, \forall i \in \{1, \ldots, mN_r\}; \ T_{i,p'}' \geq 0, \forall i, p'$$

Drawing on the Jonker-Volgenant algorithm Jonker & Volgenant (1988), this problem can be addressed as a reduced assignment problem in each step of the $K$-means algorithm, with a complexity of $O(n^3)$. The final solution provides the optimized grouping strategy for routed experts with indices:

$$S_{N_r,p} = \{i \ : \exists \ T_{i,k}' = 1, \text{ for } k \in \{m(p-1)+1, \ldots, mp\}\}$$

A.4 DETAILED ROUTER CONSTRUCTION OPTIMIZATION

This section provides the complete mathematical derivation for the router construction presented in the main manuscript.

**Problem Formulation.** Given the same input $\mathbf{x}$, the original dense FFN output equals the sum of all expert outputs: $F(\mathbf{x}) = E^s(\mathbf{x}) + \sum_{i=1}^{N_r} E_i^r(\mathbf{x})$. The MoE version differs only in the expert gating scores $\mathbf{g}$. To preserve knowledge from the original FFN, we formulate the router construction as:

$$\arg\min_G |F_{MoE}(\mathbf{x}; G) - F(\mathbf{x})| = \arg\min_G |\sum_{i=1}^{N_r} (g_i - 1) E_i^r(\mathbf{x})| = \arg\min_G |\sum^{i \in S_{de}} E_i^r(\mathbf{x})| \tag{15}$$

where $S_{de} = \{i : s_i \notin \text{TopK}(\{s_i | 1 \leq j \leq N_r\}, N_k)\}$ represents deactivated experts.

**Optimization Reduction.** Using our sparsity hypothesis from Equation (7), we reformulate the problem:

$$\arg\min_G |\sum^{i \in S_{de}} E_i^r(\mathbf{x})| \overset{\text{by } equation\ 6}{=} \arg\min_G |\sum^{i \in S_{de}} \sum_{j \in S_{N_r,i}} h_j \mathbf{w}_{down,j}|$$

$$\overset{\text{by } equation\ 7}{\approx} \arg\min_G |\sum^{i \in S_{de}} (\sum_{j \in S_{N_r,i}} |h_j|)|$$

$$= \arg\min_G \mathbb{E}_{\mathbf{h}} [\|\mathbf{h}_i^r\|_1 \mid i \in S_{de}] \tag{16}$$

Table B.1: Near-dense performance with optimized industrial settings (Llama-2 70B).

| Method | Sparsity | PIQA | WinoGrande | ARC-E | ARC-C | HellaSwag |
|---|---|---|---|---|---|---|
| Dense | 0% | 82.70 | 77.98 | 80.98 | 57.34 | 83.84 |
| Proposed (Optimized) | 25% | 82.35 | 77.41 | 80.21 | 56.50 | 83.77 |
| Degradation | | -0.35% | -0.57% | -0.77% | -0.84% | -0.07% |

**Optimal Solution via Permutation Matching.** The optimal router should match the sorting indices of expert scores $\{s_1, \ldots, s_{N_r}\}$ with expected expert activations $\{\bar{\mathbf{h}}_1^r, \ldots, \bar{\mathbf{h}}_{N_r}^r\}$ where $\bar{\mathbf{h}}_i^r = \mathbb{E}_{\mathbf{h}}[\|\mathbf{h}_i^r\|_1]$. Formally, there exists a permutation $\sigma$ such that:

$$s_{\sigma(1)} \leq s_{\sigma(2)} \leq \cdots \leq s_{\sigma(N_r)} \text{ and } \bar{\mathbf{h}}_{\sigma(1)}^r \leq \bar{\mathbf{h}}_{\sigma(2)}^r \leq \cdots \leq \bar{\mathbf{h}}_{\sigma(N_r)}^r \tag{17}$$

The minimum value of Appendix A.4 is achieved when:

$$\min_G \mathbb{E}_{\mathbf{h}}\left[\|\mathbf{h}_i^r\|_1 \mid i \in S_{de}\right] = \frac{1}{N_r - N_k} \sum_{i=1}^{N_r - N_k} \bar{\mathbf{h}}_{\sigma(i)}^r$$

**Representative Neuron Construction.** For each expert cluster, we identify the representative neuron $R_j$ as the neuron whose activation feature vector (from matrix $\mathbf{A}$) is closest to the cluster centroid:

$$R_j = \underset{i \in S_{N_r,j}}{\arg\min} \|\mathbf{c}_i - \hat{\mathbf{c}}_j\|_2 \tag{18}$$

where $\mathbf{c}_i$ are the columns of activation matrix $\mathbf{A}$ and $S_{N_r,j}$ contains the neuron indices assigned to expert $j$.

The router is constructed using these representative neurons:

$$G(\mathbf{x}) = \text{Swish}(\mathbf{x}\mathbf{W}_{gate}^R) \odot (\mathbf{x}\mathbf{W}_{up}^R) \tag{19}$$
$$= [h_{R_1}^r, h_{R_2}^r, \cdots, h_{R_{N_r}}^r] \approx [\bar{\mathbf{h}}_1^r, \bar{\mathbf{h}}_2^r, \cdots, \bar{\mathbf{h}}_{N_r}^r] \tag{20}$$

This construction provides an approximate solution to the original optimization problem by leveraging the representative neuron assumption that $h_{R_j}^r \approx \bar{\mathbf{h}}_j^r$.

# B INDUSTRIAL APPLICATION DETAILS

In this section we provide the full industrial-scale evaluation results described in the main text. These experiments use more generous calibration and inference settings (e.g., larger calibration sets, prompt engineering, self-consistency) to mimic deployment scenarios.

**Near-Dense Performance with Optimized Settings**. Table B.1 reports the framework's performance on Llama-2 70B when deployed with optimized settings. Accelerating larger models like 70B is particularly crucial for industrial deployment due to their higher computational demands. With enhanced calibration data and inference optimization techniques, the 25% sparsity configuration achieves near-dense performance across all benchmarks, with degradation typically under 1%.

**Inference Speedup for Industrial Deployment**. Table B.2 shows measured full-model speedups for the proposed method with 25% sparsity across different configurations and context lengths on Qwen-2.5 72B. A 4k context length represents typical conversational applications, while a 32k context length captures long-document processing scenarios; batch size 128 corresponds to memory-bound regimes, while larger batch sizes (BS>400) reflect compute-bound deployments. The method consistently delivers practical acceleration across both axes.

# C PERPLEXITY–SPARSITY TRADE-OFFS

Table C.1 studies WikiText-2 perplexity on Llama-2 7B as we vary FFN sparsity with a total of 16 experts. Perplexity improves monotonically as sparsity increases, and at the highest sparsity we

Table B.2: Full-model inference speedup for the proposed method with 25% sparsity across deployment scenarios (Qwen-2.5 72B). **S**: Shared experts; **A**: Active routed experts; **E**: Total experts.

| Configuration | Memory-Bound (BS=128) | | Compute-Bound (BS>400) | |
|---|---|---|---|---|
| | 4k Context | 32k Context | 4k Context | 32k Context |
| S1A5E8 | 1.08× | 1.15× | 1.12× | 1.17× |
| S3A3E8 | 1.06× | 1.13× | 1.11× | 1.148× |
| S2A4E8 | 1.05× | 1.12× | 1.10× | 1.121× |
| S4A8E16 | 1.02× | 1.10× | 1.08× | 1.11× |
| S6A6E16 | 1.03× | 1.08× | 1.07× | 1.102× |
| S3A9E16 | 1.02× | 1.05× | 1.05× | 1.085× |

Table C.1: Perplexity on WikiText-2 vs sparsity for Llama-2 7B with 16 experts. Higher sparsity corresponds to fewer active FFN parameters; lower perplexity is better.

| Sparsity | PPL-Wiki ($\downarrow$) |
|---|---|
| Dense | 5.27 |
| 0.75 | 12.73 |
| 0.625 | 9.56 |
| 0.5 | 7.71 |
| 0.375 | 6.55 |
| 0.25 | 5.78 |
| 0.125 | **5.25** |

Table D.1: Effect of $k$-sample self-consistency (voting) on academic benchmarks at 25% sparsity (S3A3E8). We report accuracy (%) on PIQA, ARC-E, ARC-C, and their average (Avg).

| Model | Method | $k$ (samples) | PIQA | ARC-E | ARC-C | Avg |
|---|---|---|---|---|---|---|
| Llama-2 7B | Dense | 1 | 78.78 | 74.58 | 46.16 | 66.51 |
| | Dense | 5 | 79.21 | 75.29 | 46.75 | 67.08 |
| | Ours (25%) | 1 | 74.34 | 67.09 | 40.35 | 60.59 |
| | Ours (25%) | 5 | 77.52 | 73.88 | 44.54 | 65.31 |
| Qwen3-30B-A3B | Dense | 1 | 84.51 | 84.43 | 57.88 | 75.61 |
| | Dense | 5 | 85.11 | 85.33 | 58.12 | 76.19 |
| | Ours (25%) | 1 | 80.23 | 76.75 | 48.80 | 68.59 |
| | Ours (25%) | 5 | 84.56 | 84.75 | 57.19 | 75.50 |

tested (0.125) our converted model slightly outperforms the dense baseline (5.25 vs 5.27), showing that aggressive activation sparsity can match or even improve language modeling quality under our analytical restructuring.

## D  EFFECT OF SELF-CONSISTENCY ON SPARSE VS DENSE MODELS

Table D.1 evaluates $k$-sample self-consistency (voting) on Llama-2 7B and Qwen3-30B-A3B at 25% sparsity (S3A3E8) over PIQA, ARC-E, and ARC-C. While increasing $k$ from 1 to 5 improves both dense and sparse models, the average accuracy gain is substantially larger for our sparse conversion (e.g., +4.72 pp vs +0.57 pp on Llama-2 7B; +6.91 pp vs +0.58 pp on Qwen3-30B-A3B), nearly closing the gap to dense under the same $k$. This supports the observation that randomness from sparse activation can be effectively averaged out via self-consistency, and that our FFN-to-MoE restructuring remains competitive once such deployment-time levers are enabled.

## E  IMPACT OF EXPERT CONFIGURATION

**Impact of Expert Configuration**. We compare different expert configurations at 25% sparsity after fine-tuning to understand the trade-offs between configuration complexity and performance.

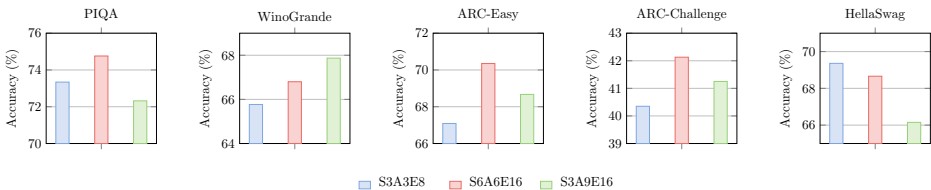

Figure E.1: Impact of expert configuration at 25% sparsity (after fine-tuning). Each subplot shows the performance of three configurations: S3A3E8 (3 shared + 3 active / 8 total), S6A6E16 (6 shared + 6 active / 16 total), and S3A9E16 (3 shared + 9 active / 16 total) across five downstream tasks.

Figure E.1 shows the performance of three configurations across five downstream tasks: S3A3E8 (3 shared + 3 active / 8 total), S6A6E16 (6 shared + 6 active / 16 total), and S3A9E16 (3 shared + 9 active / 16 total).

The results reveal interesting patterns across different tasks. S6A6E16 consistently achieves the highest performance on PIQA (74.76%) and ARC-Easy (70.35%), suggesting that balanced expert allocation with more total experts can be beneficial for knowledge-intensive tasks. However, S3A9E16 performs best on WinoGrande (67.87%), indicating that increased routing complexity can help with commonsense reasoning tasks. For ARC-Challenge, S6A6E16 again leads (42.13%), while S3A3E8 maintains competitive performance on HellaSwag (69.36%). These results demonstrate that optimal expert configuration depends on the specific downstream task characteristics, with balanced configurations generally providing robust performance across diverse evaluation scenarios.

# F DISCUSSION

**Broader Impact and Future Directions**. Our work presents an analytical post-training framework for reducing the significant computational overhead of LLM inference, thereby making powerful models more accessible for research and deployment in resource-constrained settings. Beyond a pure acceleration technique, the analytical nature of the method offers a new lens for interpreting the internal workings of FFNs. The distinct grouping of neurons into 'shared' and 'routed' experts based on activation statistics provides empirical evidence for functional specialization within these layers. Future research could leverage this methodology to analyze how knowledge is encoded and processed within LLMs. For future work, extending this analytical restructuring approach to other parts of the transformer, such as attention heads, is a promising direction. Additionally, exploring more sophisticated analytical techniques for router construction could potentially close the remaining gap with fully trained routers, without sacrificing the efficiency of the post hoc approach.

**Limitations**. While the framework provides a robust approach, its effectiveness is subject to certain conditions. Firstly, the quality of the neuron activation profiling is dependent on the calibration dataset. Performance is optimal when the calibration data is representative of the target domain, though our experiments show the method is relatively robust to calibration set size. Secondly, the discrete nature of sparse routing introduces higher variance in generation. We observe that this randomness can be effectively mitigated via self-consistency, where sparse models often benefit more from multiple samples than dense baselines.

**Compatibility with Other Efficiency Techniques**. The analytical restructuring is orthogonal to most system- and model-level efficiency methods and can be composed with them. In practice, FFN restructuring integrates well with post-training quantization (e.g., AWQ/QAT) because the operation preserves layer interfaces; it can be applied either before or after quantization with a small calibration pass to maintain accuracy. Similarly, attention-side optimizations (KV-cache compression, speculative decoding, and attention sparsity) target different bottlenecks and are complementary. Structured pruning (e.g., SliceGPT, SLEB) and our dynamic expert routing address different regimes: pruning induces static capacity reduction across all inputs, while our method activates capacity conditionally per token. Similarly, training-free activation sparsity methods (e.g., TEAL, WINA) operate at the finer neuron level and can be applied within our routed experts to further reduce FLOPs. In deployment, load-balancing and batching policies remain important to realize end-to-end speedups; our built-in bias adaptation mitigates expert hot-spotting and improves utilization on both memory-bound and compute-bound settings. Overall, the framework serves as a drop-in FFN replacement

that composes with quantization, caching, pruning, and serving optimizations to widen the practical acceleration envelope.

