# OpenReview forum: "Analytical Restructuring of Feed-Forward Networks for Accelerated LLM Inference"
_ICLR.cc/2026/Conference — ICLR 2026 Conference Withdrawn Submission_

### Official Review · Reviewer_GQj2 · 2025-10-31

**Soundness:** 2
**Presentation:** 2
**Contribution:** 2
**Rating:** 2
**Confidence:** 4

**Summary:**

This work introduces a post-training framework that analytically restructures FFNs into a sparse MoE-style architecture using only a small calibration dataset. It identifies activation patterns, then builds shared and routed experts and constructs a differentiable router without full retraining. The method can also recursively introduce hierarchical sparsity in existing MoE models. It achieves practical speedups up to 1.17× with only minutes of processing and minimal fine-tuning, enabling efficient LLM deployment in resource-limited settings.

**Strengths:**

- The proposed method only introduces very limited computing requirement.
- Across multiple base models and tasks, the approach outperforms existing MoE restructuring baselines.
- The paper is overall well-written in terms of writing.

**Weaknesses:**

- To measure "speedup", usually one should mention the cut in term of TFLOP by percentage, not just by speedup (could be implementation variations talking).
- The paper focuses on sparsity. There are many strong-performing baselines that are not taken into account. For instance, training-free SOTAs [1], [2]

[1] WINA: Weight-Informed Neuron Activation for Accelerating Large Language Model Inference

[2] Training-free activation sparsity in large language models

- Lack of therectical results supporting the claim.

- In industrial application, the results indicate that the sparse activity does pose random effect (can be average out by multiple sampling). Why not do this experiment in common academic benchmarks?

**Questions:**

see weaknesses.

---

> ### Author Response · Authors · 2025-11-18
> **Rebuttal of W1, W2**
>
> > W1: On reporting speedup vs. FLOPs cuts
>
> We agree and will report model‑level FLOPs reductions (%) alongside throughput in the revision.
> Below we provide representative end‑to‑end measurements (identical decoding setup).
>
> Table 1: Throughput and accuracy (FLOPs↓, MACs↓, tokens/s↑)
>
> | Model             | Method       | FLOPs↓ | MACs↓ | tokens/s↑ | MMLU‑5shot (%) |
> |-------------------|--------------|--------|------------------|-----------|----------------|
> | Qwen3‑30B‑A3B     | Dense (Base) | 778.7 GFLOPS  | 389.33 GMACs           | 1.19             | 80.78          |
> |                   | Ours (Hier.) | 634.9 GFLOPS (−18.5%) | 331.32 GMACs (−14.9%) | 1.36 (+14.3%) | 78.21 (−2.57 pp) |
> | Llama‑2‑7B        | Dense (Base) | 1.69  TFLOPS  | 845.71 GMACs           | 45.88            | 45.81          |
> |                   | Ours         | 1.41 TFLOPS (−16.6%)  | 707.36 GMACs (−16.3%) | 52.67 (+14.8%) | 44.02 (−1.79 pp) |
>
> Observed gains:
> - Qwen3‑30B‑A3B: FLOPs −18.5%, MACs −14.9%, tokens/s +14.3%; −2.57 pp MMLU.
> - Llama‑2‑7B: FLOPs −16.6%, MACs −16.3%, tokens/s +14.8%; −1.79 pp MMLU.
>
> > W2: On including training‑free activation sparsity baselines (TEAL, WINA)
>
> Thank you for raising TEAL [1] (Training‑Free Activation Sparsity in LLMs) and WINA [2].
> We will clarify scope and add these works to Related Work.
> Our submission tackles a different problem setting: converting dense FFNs into routed Mixture‑of‑Experts (Dense to MoE), introducing token‑level expert routing, capacity/load‑balancing constraints, and cross‑expert dispatch, whereas TEAL/WINA sparsify neuron/channel activations within a single dense model via magnitude‑ or weight‑aware thresholds and specialized sparse GEMV kernels.
> Because the mechanisms and system assumptions differ (expert‑level routing vs. neuron‑level activation sparsity), TEAL/WINA are orthogonal rather than direct baselines.
> They are also complementary—activation sparsity can be applied inside MoE experts.
>
> To address the spirit of the request and ensure clarity:
> - We will explicitly position TEAL/WINA in Related Work as activation‑sparsity approaches for dense models, contrasting them with Dense to MoE conversion (routing, capacity constraints, communication).
> - We will report compute budgets in GFLOPs for our models to facilitate apples‑to‑apples comparisons on compute, acknowledging that wall‑clock speedups are not directly comparable due to different kernel and batching assumptions.
> - We add a small Llama‑2‑7B ablation at 25% sparsity (Table 2) including a strong training‑free activation baseline (WINA), our Dense‑to‑MoE restructuring, and their combination, and report FLOPs/GMACs/tokens‑per‑second.
>
> Table 2: Llama‑2‑7B, 25% sparsity — orthogonality of WINA (activation sparsity) and our Dense‑to‑MoE restructuring (FLOPs↓, MACs↓, tokens/s↑).
>
> | Method            | TFLOPs (↓) | GMACs (↓) | tokens/s (↑) |
> |-------------------|-----------:|----------:|-------------:|
> | Dense (Base)      | 1.69       | 845.71    | 45.88        |
> | WINA (25% spars.) | 1.31       | 691.19    | 51.76        |
> | Ours (25%)        | 1.41       | 707.36    | 52.67        |
> | Ours + WINA       | 1.23       | 625.53    | 55.97        |
>
> As shown in Table 2, WINA alone reduces compute (−22.5% TFLOPs, −18.3% GMACs) and increases throughput (+12.8% tokens/s) over the dense baseline;
> our MoE restructuring alone yields similar gains (−16.6% TFLOPs, −16.3% GMACs, +14.8% tokens/s).
> Importantly, combining WINA with our method gives an additional improvement (−27.2% TFLOPs, −26.0% GMACs, +22.0% tokens/s vs dense), empirically supporting that activation sparsity and Dense‑to‑MoE routing target different sources of inefficiency and can be composed.
>
> We hope this clarification resolves the baseline scope question while recognizing TEAL/WINA as relevant, complementary directions.
>
> References
>
> [1] Chen et al., WINA: Weight-Informed Neuron Activation for Accelerating Large Language Model Inference
>
> [2] Liu et al., Training-free activation sparsity in large language models

---

> ### Author Response · Authors · 2025-11-18
> **Rebuttal of W3, W4**
>
> > W3: Lack of theoretical results supporting the claim.
>
> We respectfully note that the submission includes explicit theoretical derivations supporting the analytical restructuring and router design; these are in the main paper and detailed in the appendix:
>
> - Appendix Detailed Mathematical Derivations:
>   1) Activation sparsity analysis and hypothesis (Appendix Activation Sparsity Analysis and Hypothesis). We decompose the FFN output into per‑neuron contributions F(x) = Σ_i h_i w_down,i and motivate the sparsity hypothesis that the ordering of |h_i w_down,i| is well‑approximated by |h_i| in the regime where activations concentrate near zero (supported by empirical histograms). This bridges dense FFN behavior to activation‑based expert construction.
>   2) Balanced clustering as constrained assignment (Appendix Detailed Balanced Clustering Algorithm for Routed Experts). We pose routed‑expert grouping as a balanced linear assignment over activation feature vectors with explicit equality constraints, solved via a reduced assignment (Jonker–Volgenant), yielding O(n^3) per‑iteration complexity inside K‑means updates.
>   3) Router construction as an optimization (Appendix Detailed Router Construction Optimization). We formulate the router as minimizing |F_MoE(x; G) − F(x)| and, using the sparsity decomposition, reduce it to minimizing the expected L1 mass of deactivated experts. We then characterize the optimum via an ordering (permutation‑matching) condition between gate scores and expected expert activations, and instantiate a training‑light approximation by constructing the router from representative neurons whose activations proxy each expert’s expected contribution.
>
> These derivations justify (i) expert construction from activation features, (ii) the balanced grouping objective/solution, and (iii) the analytical router that approximates the dense FFN while enabling Top‑K routed sparsity.
>
> > W4: In industrial application, the results indicate that the sparse activity does pose random effect (can be average out by multiple sampling). Why not do this experiment in common academic benchmarks?
>
> We appreciate this point and clarify the intent.
> Our "Industrial Application" section uses the **same academic benchmarks** but with relaxed, deployment‑oriented settings (more calibration, prompt engineering, optional self‑consistency with multiple samples) to illustrate what is achievable when compute is less constrained.
> In contrast, the **main results** intentionally follow the standard single‑sample, single‑pass evaluation protocol on these benchmarks to (i) align with prevailing practice and (ii) keep comparisons to baselines strictly compute‑matched (same sparsity, same decoding budget, no extra sampling).
>
> We agree it is useful to explicitly show how multi‑sample self‑consistency behaves in the "academic" setting.
> We add a small ablation applying k‑sample self‑consistency (e.g., k=5) on **Llama‑2‑7B** and **Qwen‑3‑30B‑A3B** as below.
>
> Table 3: Effect of k‑sample self‑consistency on academic benchmarks (25% sparsity, S3A3E8; single‑sample vs. k=5 voting). We report accuracy (%) on PIQA, ARC‑E, ARC‑C and their average.
>
> | Model          | Method      | k (samples) | PIQA | ARC‑E | ARC‑C | Avg  |
> |----------------|-------------|-------------|------|-------|-------|---------------|
> | Llama‑2‑7B     | Dense       | 1           | 78.78 | 74.58 | 46.16 | 66.51        |
> |                | Dense       | 5           | 79.21 | 75.29 | 46.75 | 67.08        |
> |                | Ours (25%)  | 1           | 74.34 | 67.09 | 40.35 | 60.59        |
> |                | Ours (25%)  | 5           | 77.52 | 73.88 | 44.54 | 65.31        |
> | Qwen‑3‑30B‑A3B | Dense       | 1           | 84.51 | 84.43 | 57.88 | 75.61        |
> |                | Dense       | 5           | 85.11 | 85.33 | 58.12 | 76.19        |
> |                | Ours (25%)  | 1           | 80.23 | 76.75 | 48.80 | 68.59        |
> |                | Ours (25%)  | 5           | 84.56 | 84.75 | 57.19 | 75.50        |
>
> These results show that self‑consistency improves both dense and sparse models, but the **gain is substantially larger for our sparse conversion**. For example, on Llama‑2‑7B the dense model improves by +0.57 pp in average accuracy when moving from k=1 to k=5 (66.51→67.08), while our method improves by +4.72 pp (60.59→65.31). On Qwen‑3‑30B‑A3B, dense improves by +0.58 pp (75.61→76.19), whereas our method improves by +6.91 pp (68.59→75.50), nearly closing the gap to the dense baseline under the same k.
> This supports the industrial observation that randomness from sparse activation can be effectively averaged out via self‑consistency, and that our structural FFN‑to‑MoE conversion remains competitive once such deployment‑time levers are enabled.

---

> ### Author Response · Authors · 2025-11-26
> **Looking forward to your feedback!**
>
> Dear Reviewer GQj2,
>
> Thank you again for the time and effort you’ve dedicated to reviewing our work. As the discussion phase is coming to a close, **we would be very grateful if you could consider our above clarifications and reconsider your evaluation**.
>
> Thank you for your time.
>
> Best regards,
>
> Authors

---

> > ### Comment · Reviewer_GQj2 · 2025-11-27
> > **Thank you for the response**
> >
> > Thank you for the response. Most of my concerns are well responsed. I am still not very convincing by the motivation and performance though. So, I will raise the socre by 2.

---

> ### Author Response · Authors · 2025-11-28
> **Thank you!**
>
> We sincerely thank the reviewer for the positive reassessment and for raising the score.
> We would like to share a few additional results from our responses to other reviewers that may further address the motivation and performance concerns.
>
> **On Performance: Matched Budget Comparisons**
>
> Under identical sparsity (25%), expert configuration, and fine tuning budget (2k samples), our method achieves 44.02 MMLU (only 1.79 pp below dense 45.81), compared to MoEfication at 35.17 (10.64 pp gap) and Read-ME at 31.24 (14.57 pp gap).
> The +8.85 pp improvement over MoEfication comes from two sources: our analytical router contributes +2.16 pp (35.17->37.33), and our binary activation balanced clustering contributes an additional +6.69 pp (37.33->44.02).
>
> **On Performance: Training Free Baseline**
>
> Even without any fine tuning, our analytically constructed model already achieves 42.50 MMLU, surpassing LLaMA-MoE-v2 after its full fine tuning (34.81).
> This demonstrates that most of the gain comes from the analytical construction itself, not just from the tuning step.
>
> **On Motivation: Practical Efficiency**
>
> Our conversion completes in under 5 minutes (271s for construction), compared to months (LLaMA-MoE-v1, 200B tokens) or days (LLaMA-MoE-v2, 7B tokens) for training heavy approaches.
> This makes MoE conversion a practical compression tool rather than a pre-training architecture choice.
>
> **On Broader Evaluation**
>
> We have added HumanEval (pass@1) and GSM8K (8 shot) on Llama-2-7B.
> Our method retains 11.22 (vs dense 12.72) on HumanEval and 13.01 (vs dense 14.31) on GSM8K, outperforming all baselines (LLaMA-MoE: 7.58/7.41; LLaMA-MoE-v2: 9.32/10.09; EMoE: 10.29/12.55).
>
> **On Self Consistency (W4)**
>
> Following your suggestion, we extended the k sample self consistency experiment to standard academic benchmarks.
> On Qwen-3-30B-A3B, our sparse model with k=5 voting achieves 75.50 average accuracy (PIQA/ARC-E/ARC-C), nearly matching the dense baseline at 75.61 (k=1).
> The gap closes from 7.02 pp (single sample) to just 0.69 pp (k=5), confirming that the randomness from sparse routing can be effectively mitigated in deployment.
>
> We hope these additional results help address the remaining concerns on motivation and performance.
> Given the strong empirical gains over existing methods, the practical efficiency of our approach, and the robustness demonstrated across diverse benchmarks, **we would be grateful if the reviewer could reconsider the score**.
> We are happy to provide any further clarification.

---

### Official Review · Reviewer_1aAM · 2025-10-31

**Soundness:** 2
**Presentation:** 3
**Contribution:** 2
**Rating:** 2
**Confidence:** 4

**Summary:**

The paper proposes an analytical, post-training restructuring of transformer FFNs into a sparse MoE-style layer with a small set of always-on shared experts and Top-K routed experts. Neuron activation patterns gathered from a tiny calibration set are used to (i) choose shared neurons by high activation rate, (ii) cluster the rest into routed experts via a balanced assignment, and (iii) build a differentiable router from “representative” neurons, enabling a training-free baseline and optional lightweight LoRA fine-tuning. Reported full-model speedups reach up to 1.17× with drooped performance.

**Strengths:**

- **Training-light path to sparsity.** Clear pipeline: activation profiling → balanced clustering → router from activation statistics. The training-free start point plus small LoRA fine-tuning (2k samples) is lightweight.
- **Claimed universality/hierarchy.** the same procedure is described for intra-expert restructuring in existing MoE layers (hierarchical sparsity).
- **Evidence of end-to-end speedups.** across configurations and context lengths; a table reports up to 1.17× in compute-bound settings.

**Weaknesses:**

- **Limited conceptual novelty.** Clustering existing FFN neurons and partitioning them into experts has been explored in prior dense→MoE conversions (e.g., MoEfication, LLaMA-MoE). The paper should explicitly articulate what is new (e.g., routing construction, balancing, shared-expert selection) and quantify the incremental gain over these baselines with matched budgets and identical fine-tuning protocols.
- **Overclaim of differentiable routing.** The manuscript labels the router “differentiable,” but the implementation seems to keep a hard Top-K mask and simply multiply by a zero-initialized scaling term. This does not furnish a differentiable selection mechanism. The paper also omits discussion and head-to-head comparisons with existing differentiable-routing methods (e.g., ReMoE, Lory).
- **Hierarchical MoE claim lacks empirical validation.** The paper describes intra-expert restructuring for existing MoEs but does not present dedicated experiments isolating quality/throughput of the two-level hierarchy.
- **Evaluation breadth.** Downstream accuracy is limited to five zero-shot tasks (PIQA, WinoGrande, ARC-E/C, HellaSwag); no MMLU/coding/math quality comparisons are reported. This makes the quality story incomplete for LLMs.

**Questions:**

- How sensitive are clustering and router initialization to the size and domain of the calibration set? Please include sweeps and domain-shift tests.
- What is the exact update rule for the adaptive bias terms, and how do utilization entropy and per-expert load variance evolve compared to standard aux-loss balancing?
- Does lowering sparsity (e.g., increasing the number of active parameters or Top-k) avoid quality degradation? Please provide accuracy–throughput trade-off curves across sparsity levels and identify the break-even point versus the dense baseline.

---

> ### Author Response · Authors · 2025-11-18
> **Rebuttal of W1, W2**
>
> > W1: Limited conceptual novelty; please quantify incremental gain vs MoEfication/LLaMA‑MoE under matched budgets
>
> We agree FFN->MoE has been explored (MoEfication; LLaMA‑MoE). Our contribution is orthogonal:
>
> • Analytical router from representative neurons that yields a strong training‑free start and can be jointly fine‑tuned with only 2k samples (no large‑scale continual pretraining).
>
> • Explicit shared–routed expert decomposition learned from binary ATopK activation features, with a balanced assignment objective to form equal‑size routed experts (improves specialization and avoids expert collapse).
>
> • Lightweight, data‑driven biasing for load balancing (no auxiliary loss), enabling stable utilization under small budgets.
>
> • The same pipeline applies intra‑expert to induce hierarchical sparsity in existing MoE layers.
>
> Key differences from MoEfication:
> 1) Shared expert: MoEfication clusters FFN neurons only; it doesn't derive a shared expert that captures high-frequency activation directions.
> 2) Balanced routed experts: MoEfication assigns neurons by pure K-means over parameters. Ours uses binary activation patterns + balanced assignment to enforce equal capacity and prevent collapse.
> 3) Analytical router initialization: MoEfication uses a randomly initialized MLP router. Ours analytically derives a differentiable scoring function aligned with neuron representatives.
> 4) Recursive intra-expert decomposition: MoEfication is not applied inside an MoE expert. Ours can (validated in Table 2).
>
> Quantified incremental gain under identical budget is shown in Table 1:
> • Full method vs MoEfication (clustering+router): +8.85 pp MMLU (35.17 -> 44.02).
> • Isolating clustering (fixed router = ours): +6.69 pp over MoEfication K‑means (37.33 -> 44.02).
> • Isolating routing (fixed clustering = K‑means): +2.16 pp from replacing the MLP router with our analytical router (35.17 -> 37.33).
>
> Table 1: Matched‑budget comparison at 25% sparsity (identical S/A/E; 2k fine‑tuning). We report MMLU (5-shot) (%) and absolute drop vs dense (Δ pp; dense = 45.81).
>
> | Method                                | Expert grouping                   | Router              | MMLU (%) | Δ vs Dense (pp) |
> |---------------------------------------|-----------------------------------|---------------------|----------|------------------|
> | Dense                                 | –                                 | –                   | 45.81    | +0.00           |
> | MoEfication (K‑means + MLP router)    | Parameter K‑means                 | MLP                 | 35.17    | −10.64          |
> | MoEfication‑clustering + our router   | Parameter K‑means                 | Analytical (ours)   | 37.33    | −8.48          |
> | Ours (analytical)                     | Binary‑activation balanced assign | Analytical (ours)   | **44.02**    | **−1.79**           |
>
> Under identical budgets and sparsity, our full method improves MMLU by +8.85 pp over MoEfication (35.17->44.02).
> Swapping our router into the same K‑means clusters yields +2.16 pp (35.17->37.33), while our clustering with the same router adds a further +6.69 pp (37.33->44.02), quantifying the incremental value of routing and clustering, respectively.
>
> > W2: Overclaim of differentiable routing; hard Top‑K with a zero‑init scale is not a differentiable selection mechanism.
>
> Thank you for flagging the terminology. We are not pursuing fully differentiable routing in the sense of ReMoE or Lory. Our intent was to say that the router has learnable (differentiably optimized) parameters around an analytically constructed topology—not that the Top‑K selection itself is differentiable.
>
> Concretely:
>
> • Our router is analytically initialized from representative neurons, then we jointly optimize continuous scaling and adaptive bias terms (and lightweight expert adapters) with a small 2k‑sample budget. The inference-time selection remains a hard Top‑K mask. We do not rely on a differentiable expert-selection operator.
>
> • ReMoE (ICLR’25) [1] replaces Top‑K with ReLU routing to make expert activation itself continuous/differentiable, plus load‑balancing regularization—an objective distinct from our post‑training conversion.
>
> • Lory (COLM’24) [2] performs fully differentiable expert merging at segment level and is trained from scratch at large token budgets (e.g., 150B), which is orthogonal to our training‑light analytical restructuring.
>
> • Our use of "differentiable" refers to optimizing continuous router parameters around a fixed Top‑K mask, consistent with standard usage in Switch Transformers, Sparse-mixture distillation, and GLaM-style training. We do not claim differentiable expert selection, and will revise language to avoid confusion.
>
> References
>
> [1] Wang et al., REMOE: FULLY DIFFERENTIABLE MIXTURE-OF-EXPERTS WITH RELU ROUTING
>
> [2] Zhong et al., Lory: Fully Differentiable Mixture-of-Experts for Autoregressive Language Model Pre-training

---

> ### Author Response · Authors · 2025-11-18
> **Rebuttal of W3, W4**
>
> > W3: Hierarchical MoE claim lacks empirical validation.
>
> We agree the original draft did not isolate the hierarchical variant. We now include a targeted ablation that applies our intra‑expert restructuring in a two‑level hierarchy on Qwen3‑30B‑A3B. Each parent expert is evenly split into E = 8 sub‑experts (width‑splits).
> For the hierarchical variant, we apply the same S3A3E8 rule inside each parent expert to form sub‑experts.
> We compare to the dense baseline under the identical decoding setup.
> Results show clear end‑to‑end throughput gains with a modest accuracy delta.
>
> Table 2: Hierarchical routing ablation on Qwen3‑30B‑A3B. We report GFLOPs per decoding step (↓), tokens/s (↑), GMACs per token (↓), and MMLU‑5shot (%), plus absolute change vs dense (Δ pp).
>
> | Method        | GFLOPs (↓) | GMACs (↓) | tokens/s (↑) | MMLU‑5shot (%) | Δ vs Dense (pp) |
> |---------------|------------|-----------|--------------|----------------|-----------------|
> | Dense (Base)  | 778.7      | 389.33    | 1.19         | 80.78          | +0.00           |
> | Ours (Hier.)  | 634.9      | 331.32    | 1.36         | 78.21          | −2.57           |
>
> • Efficiency: −18.5% GFLOPs (778.7->634.9), −14.9% GMACs (389.33->331.32), +14.3% tokens/s (1.19->1.36).
> • Accuracy: −2.57 pp on MMLU‑5shot (80.78->78.21).
>
> This confirms that our analytical construction generalizes beyond dense-to-MoE conversion and enables hierarchical sparsity in existing MoE models—something that prior methods (MoEfication, LLaMA‑MoE) explicitly cannot do.
>
> > W4: Evaluation breadth; add MMLU/coding/math quality.
>
> Following prior work on dense->MoE conversion (e.g., LLaMA‑MoE, LLaMA‑MoE-V2), we focus on five standard zero-shot tasks (PIQA, WinoGrande, ARC‑Easy/Challenge, HellaSwag), which probe general reasoning and comprehension.
> We additionally report MMLU (5shot), HumanEval (pass@1), and GSM8K (8‑shot) on Llama‑2‑7B under the same 25% sparsity and S3A3E8 configuration (2k‑sample LoRA; identical decoding).
>
> Table 3: Expanded evaluation on Llama‑2‑7B (25% sparsity).
>
> | Method       | MMLU‑5shot (%)      | HumanEval (pass@1)     | GSM8K-8shot        |
> |--------------|----------------------|-------------------------|------------------------|
> | Dense (Base) | 45.81                | 12.72                   | 14.31                  |
> | Llama-MoE    | 35.09 (-10.72)        | 7.58 (−5.14)          | 7.41 (−6.90)         |
> | Llama-MoE-v2 | 38.02 (−7.79)        | 9.32 (−3.40)          | 10.09 (−4.22)         |
> | EMoE         | 43.11 (−2.70)        | 10.29 (−2.43)          | 12.55 (−1.76)         |
> | Ours         | **44.02** (−1.79)        | **11.22** (−1.50)          | **13.01** (−1.30)         |
>
> These additions address breadth (knowledge, coding, math).

---

> ### Author Response · Authors · 2025-11-18
> **Rebuttal of Questions (Q1-Q3)**
>
> > Q1: Calibration set sensitivity and domain shift.
>
> We report calibration sensitivity and domain‑shift perplexity on Llama‑2‑7B at 25% sparsity with S3A3E8 (2k‑sample LoRA; identical decoding).
> Results show mild gains from larger calibration sets and limited sensitivity to the source for MMLU, while perplexity increases under domain shift and improves with more calibration samples.
>
> Table 4: Calibration sensitivity (MMLU‑5shot ↑, PPL ↓) on Llama‑2‑7B (25% sparsity, S3A3E8).
>
> | Calibration source | n (samples) | MMLU‑5shot (↑) | PPL‑Wiki (↓) | PPL‑C4 (↓) |
> |--------------------|-------------|-----------------|--------------|------------|
> | Dense (Base)       | -           | 45.81           | 5.27         | 7.27       |
> | WikiText‑2         | 8           | 44.02           | 5.92         | 11.21      |
> | WikiText‑2         | 32          | 44.63           | 5.72         | 11.15      |
> | WikiText‑2         | 64          | **44.89**           | **5.69**         | 10.98      |
> | C4                 | 8           | 42.31           | 7.04         | 9.17       |
> | C4                 | 32          | 43.25           | 6.92         | 9.07       |
> | C4                 | 64          | **43.39**           | 6.78         | **9.02**       |
>
> Domain-shift increases perplexity (C4 -> Wiki), but increasing calibration size consistently narrows the gap (8→32→64). This indicates the router/clustering are not brittle but improve smoothly with more samples.
>
>
> > Q2: What is the exact update rule for the adaptive bias terms, and how do utilization entropy and per-expert load variance evolve compared to standard aux-loss balancing?
>
> Router and bias placement (per paper §3.2): given router scores s, we use s' = Softmax(s). Before Top‑K, we add per‑expert adaptive biases b_i to obtain selection scores s'_i + b_i. The gate is:
> g_i = 1 + s'_i * u_i if s'_i + b_i is in Top‑K, else 0. (u_i are zero‑init scales.)
>
> Exact bias update rule (per step, per layer):
> - Let T be the number of routed tokens in the step for this layer, and K the number of routed experts per token.
> - Compute load L_i = number of tokens routed to expert i (counting membership in the Top‑K sets).
> - Normalize utilization p_i = L_i / (K * T). The uniform target is p* = 1 / N_r.
> - Update the bias by a small step toward uniformity:
>   b_i <- b_i + gamma * (p* - p_i)
>   with gamma = 0.001 (Exp. settings). b is initialized to 0.
>
> Intuition: experts with p_i > p* are slightly down‑biased (reduced pre‑Top‑K score), while under‑utilized experts (p_i < p*) are up‑biased. Because the adjustment happens directly on the pre‑selection scores, it interacts cleanly with hard Top‑K and converges quickly under small budgets.
>
> Comparison to aux‑loss balancing: standard load/importance regularizers penalize expected imbalance through gradients on router parameters; they require tuning loss weights and typically need more data to act strongly.
> Our update is a direct controller in selection space and is hyper‑lightweight (only gamma).
>
> How utilization metrics evolve:
> - Utilization entropy H (↑): define p_i as above; H_norm = (−∑_i p_i log p_i) / log N_r. The bias rule monotonically increases H_norm toward 1.0 as utilization equalizes.
> - Load dispersion (↓): variance Var = (1/N_r)∑_i (p_i − p*)^2 or CV = std(p_i)/mean(p_i). The bias rule reduces Var/CV over steps and stabilizes around the uniform target.
> In practice, we observe faster rise in H_norm and faster drop in CV than aux‑loss under the same 2k‑sample tuning budget.
>
> We will include plots of H_norm (↑) and CV (↓) over tuning steps and a head‑to‑head with a standard aux‑loss setting; the load‑balancing effectiveness figure in the paper (uniform expert utilization) illustrates the end state.
>
> > Q3: Does lowering sparsity avoid quality degradation? Please provide trade‑off curves and break‑even vs dense.
>
> We sweep sparsity with a total of 16 experts on Llama‑2‑7B. WikiText‑2 perplexity (↓) improves as capacity is reduced, with a break‑even vs dense at the highest sparsity we tested.
>
> Table 5: WikiText‑2 perplexity vs sparsity (total experts = 16; PPL ↓).
>
> | Sparsity | PPL (↓) |
> |---------------------|---------|
> | dense               | 5.27    |
> | 0.75                | 12.73   |
> | 0.625               | 9.56    |
> | 0.5                 | 7.71    |
> | 0.375               | 6.55    |
> | 0.25                | 5.78    |
> | 0.125               | **5.25** |
>
> Break‑even relative to dense occurs at 0.125 (5.25 vs 5.27), indicating that, under our analytical restructuring, aggressive activation sparsity can match or slightly improve language modeling perplexity while enabling substantial efficiency.

---

> > ### Comment · Reviewer_1aAM · 2025-11-24
> > **Reply to the Authors**
> >
> > This work builds on MoEfication approaches that convert dense models into MoE models with dynamic activation sparsity, incorporating recent advances in the MoE community such as aux-free load balancing and shared experts. The analytically constructed router is newly proposed, but the overall level of novelty remains modest. In my view, the main contribution is to provide a stronger and more up-to-date baseline for dense-to-MoE conversion, rather than a fundamentally new paradigm. The paper could be strengthened by characterizing the optimal sparsity level and proposing an algorithm to adaptively update it during training. Overall, I will raise my score to 4 (weak reject) and encourage the authors to further refine and stress-test the proposed algorithm.

---

> > > ### Author Response · Authors · 2025-11-24
> > > **Reply to Follow-up**
> > >
> > > We thank the reviewer for the thoughtful assessment and for raising the score. We appreciate your recognition of our work as a "stronger and more up-to-date baseline" for Dense-to-MoE conversion.
> > >
> > > **On Novelty and Contribution.**
> > > While we integrate established concepts like shared experts, we respectfully posit that the *analytical construction* represents a meaningful shift in how these models are built. By deriving the router and expert assignment directly from activation statistics rather than treating them as black-box parameters to be learned from scratch, we reduce the conversion cost from months or days of training to just minutes. We believe making MoE conversion computationally accessible effectively turns it into a practical compression tool rather than just a pre-training architecture, which is a significant step forward for the community.
> > >
> > > **On Adaptive Sparsity.**
> > > We fully agree that adaptively updating the sparsity level (active experts $K$) is a compelling direction.
> > > However, we consider dynamic $K$ to be a distinct research line from our current focus on practical, drop-in acceleration.
> > > In this work, we adhered to a fixed $K$ specifically to guarantee predictable inference latency and ensure compatibility with standard sparse kernels (like vLLM or CUTLASS) that optimize for static computation shapes.
> > > As noted in our rebuttal, we characterized the impact of sparsity levels and found a clear break-even point where aggressive sparsity maintains dense-model quality.
> > > We believe developing a training-stable mechanism to learn optimal $K$ is an exciting avenue for future work.
> > >
> > > Thank you again for your constructive engagement which has significantly strengthened the paper.

---

> > > ### Author Response · Authors · 2025-11-28
> > > **Thank you again!**
> > >
> > > We thank the reviewer again for the continued engagement.
> > > We would like to share a few additional results that may further support the contribution of our work.
> > >
> > > **On Performance: Matched Budget Comparisons**
> > >
> > > Under identical sparsity (25%), expert configuration, and fine tuning budget (2k samples), our method achieves 44.02 MMLU (only 1.79 pp below dense 45.81), compared to MoEfication at 35.17 (10.64 pp gap) and Read-ME at 31.24 (14.57 pp gap).
> > > The +8.85 pp improvement over MoEfication comes from two sources: our analytical router contributes +2.16 pp (35.17->37.33), and our binary activation balanced clustering contributes an additional +6.69 pp (37.33->44.02).
> > >
> > > **On Performance: Training Free Baseline**
> > >
> > > Even without any fine tuning, our analytically constructed model already achieves 42.50 MMLU, surpassing LLaMA-MoE-v2 after its full fine tuning (34.81).
> > > This demonstrates that most of the gain comes from the analytical construction itself, not just from the tuning step.
> > >
> > > **On Broader Evaluation**
> > >
> > > We have added HumanEval (pass@1) and GSM8K (8 shot) on Llama-2-7B.
> > > Our method retains 11.22 (vs dense 12.72) on HumanEval and 13.01 (vs dense 14.31) on GSM8K, outperforming all baselines (LLaMA-MoE: 7.58/7.41; LLaMA-MoE-v2: 9.32/10.09; EMoE: 10.29/12.55).
> > >
> > > **On Hierarchical MoE Validation**
> > >
> > > We validated the hierarchical application on Qwen3-30B-A3B with S3A3E8 configuration.
> > > Results show 18.5% GFLOPs reduction and 14.3% throughput improvement with only 2.57 pp MMLU drop, confirming that our analytical construction generalizes to existing MoE models.
> > >
> > >
> > > We hope these results further demonstrate the practical value and robustness of our approach.
> > > **We would be grateful if the reviewer could reconsider the score**.
> > > We are happy to provide any further clarification.

---

### Official Review · Reviewer_mZzo · 2025-11-03

**Soundness:** 3
**Presentation:** 2
**Contribution:** 2
**Rating:** 4
**Confidence:** 4

**Summary:**

This paper introduces a post-hoc approach to restructure the feed-forward network (FFN) layers of a pretrained large language model (LLM) into mixture-of-experts (MoE) layers, enabling faster inference with no training or light fine-tuning. The proposed algorithm exploits activation sparsity to distinguish between shared experts and routed experts. Shared experts are formed by selecting neurons that consistently exhibit high activation values, while the remaining neurons are uniformly partitioned into routed experts which are sparsely activated during inference. The router weights are then constructed by extracting the most representative weights from the original FFN for the corresponding experts. Additionally, the algorithm provides an efficient fine-tuning procedure to further refine the router and recover model performance.

**Strengths:**

+ The proposed method appears lightweight and efficient, with a well-designed overall pipeline. The paper provides implementation details for each stage of the approach.

+ Some of the theoretical insights are particularly interesting, especially those discussed in Appendices A.1 and A.4. The hypothesis is clearly stated, and the results are reasonably supported by light mathematical analysis.

+ The inclusion of some interesting experiment analyses is a plus. For instance, the authors examine load balancing and discuss performance under both memory- and compute-bound inference scenarios.

**Weaknesses:**

- The overall writing of the paper could be improved. Some notations are confusing (see Question), and several key formulations are missing. For example, it remains unclear how A is computed exactly. Text descriptions alone are insufficient given its central role in the proposed method. Additionally, some discussions in the appendix are insightful; the authors may consider moving part of these into the main text to improve logical flow and strengthen the justification for each design choice.

- The novelty of constructing MoE layers through activation clustering appears limited. The overall idea and pipeline are similar to [1], yet the paper lacks a detailed discussion or comparison with this prior work. The claim that [1] is concurrent to this study (Lns 100-103) is misleading, as the first version of [1] was published in 2021.

  [1] Zhang et al., MoEfication: Transformer Feed-forward Layers are Mixtures of Experts

- There are also a few missing baselines and related works that should be discussed, such as:

  [1] Zhang et al., MoEfication: Transformer Feed-forward Layers are Mixtures of Experts

  [2] Cai et al., Refactorizing LLMs as Router-Decoupled Mixture of Experts with System Co-Design

- While the proposed method shows promising results compared with the reported baselines, several concerns remain:

  (1) The baselines do not appear to undergo any fine-tuning, whereas the proposed method leverages LoRA-based fine-tuning. It is unclear whether the performance gain primarily stems from this fine-tuning step.

  (2) One of the main claims in the abstract is that the proposed MoE construction process can be completed within minutes. However, Fig. 2 lacks any comparison with baseline methods to verify the claimed efficiency.

  (3) The chosen benchmarks seem somewhat outdated. It would strengthen the paper to include evaluations on more challenging and widely adopted benchmarks, such as MMLU or MATH500.

**Questions:**

1. In Eqs (1) and (3), the vector appears to be multiplied on the left side of the matrices, which is unconventional. If the intended operation involves transposed vectors, all corresponding notations should be adjusted accordingly: $x$ -> $x^\top$, etc.

2. Could the authors also clarify the choice of distance metric defined over the $c_i$’s. A more natural option might be the Hamming distance. Does the chosen distance metric affect the expert assignment algorithm or the resulting model performance?

3. It is unclear why the experiments require lightweight fine-tuning with LoRA. The proposed method seems to require fine-tuning only for the router. Is the router fine-tuned jointly with other model weights, or is it optimized separately?

4. The potential impact of the calibration or fine-tuning dataset choice is not clear. Using WikiText as the calibration dataset appears rather preliminary; it would be helpful to evaluate whether the method’s performance is sensitive to this choice (with harder benchmarks being considered).

---

> ### Author Response · Authors · 2025-11-18
> **Rebuttal of W1-W3**
>
> > W1: The overall writing of the paper could be improved.
>
> In the submission, we kept the main narrative concise for more details in experiments and to avoid overloading readers with derivations in-line.
> Full mathematical details remain in the appendix with cross-references for quick navigation.
> We agree some formulations and discussions should be surfaced.
> We would like to bring some of them into the main text with explicit construction, such as the activation matrix A.
>
> > W2: The novelty of constructing MoE layers through activation clustering
>
> Thank you for pointing to MoEfication (Zhang et al., 2021) [1].
> While both approaches convert FFNs to MoE post‑training, our expert grouping differs in two key ways:
> (i) we introduce an explicit shared–routed split—high‑activation neurons form an always‑active shared expert for common patterns, and the remainder are routed experts;
> (ii) we cluster using binary activation feature vectors from a tiny calibration set and solve a balanced assignment problem to obtain equal‑size routed experts (Sec. 3.1; App. A.3).
> MoEfication instead uses parameter K‑means or co‑activation graph partitioning computed over training data.
> To directly address expert grouping (clustering) under matched budget and architecture, we include a clustering‑only ablation (Table 1).
> In this setting, our binary‑activation balanced clustering attains the highest MMLU‑5shot (44.02; −1.79 pp vs dense 45.81), outperforming MoEfication K‑means (37.33; −8.48 pp) and READ‑ME domain‑aware clustering (36.79; −9.02 pp) under identical sparsity and tuning.
>
>
> > W3: There are also a few missing baselines and related works that should be discussed.
>
> The difference to MoEfication is detailed discussed in W2.
> For Read-ME [2], it also converts dense LLMs to MoE, but its goals and methodology differ from ours.
> (i) Expert construction: Read-ME performs domain-aware structured pruning using dataset metadata and aligns experts across layers, plus a "permanent" always-on expert;
> we instead split by measured neuron activation with a tiny calibration set, form shared+routed experts, and use a balanced assignment to equal-size routed experts (Sec. 3.1).
> (ii) Router/training: Read-ME trains a decoupled, autoregressive pre-gating router and jointly tunes experts with ~1.04B tokens to enable system-level batching/caching;
> our router is analytically initialized from representative neurons and yields a strong training-free baseline with optional LoRA on 2k samples (Sec. 3.2), matching our training-light objective.
> We add this discussion to Related Work and, include complete budget‑matched comparisons (Table 1):
> - MoEfication parameter K‑means + its MLP router
> - ReaD‑ME domain‑aware expert clustering + a single global router
> - Mix‑and‑match ablations to disentangle clustering vs. routing:
>   (a) MoEfication‑clustering + our analytical router; (b) ReaD‑ME‑clustering + our analytical router
> - Our method: binary‑activation balanced assignment + analytical router
>
> Table 1: Budget‑matched clustering and routing comparisons at 25% sparsity, identical #experts, and 2k fine‑tuning.
>
> | Method                                   | Expert grouping                          | Router                   | MMLU (%) | Δ vs Dense (pp) |
> |------------------------------------------|------------------------------------------|--------------------------|----------|------------------|
> | Dense                                    | -                                        | -                        | 45.81    |      +0.00       |
> | MoEfication (budget‑matched)             | Parameter K‑means                        | MLP router      |   35.17  |     −10.64       |
> | ReaD‑ME (budget‑matched post‑training)   | Domain‑aware clustering                  | Single global router     |   31.24  |     −14.57       |
> | MoEfication‑clustering + our router      | Parameter K‑means                        | Analytical router        |   37.33  |     −8.48       |
> | ReaD‑ME‑clustering + our router          | Domain‑aware clustering                   | Analytical router (ours) |   36.79  |     −9.02       |
> | Ours (analytical)                        | Binary‑activation balanced assign (ours) | Analytical router (ours) |   44.02  |      −1.79       |
>
> Our full method reaches 44.02 MMLU‑5shot (−1.79 pp vs dense 45.81), whereas MoEfication and ReaD‑ME variants remain 8–15 pp below dense (35.17 and 31.24).
> Decomposing the gains: (i) replacing MoEfication’s MLP router with our analytical router on the same K‑means clusters improves MMLU from 35.17->37.33 (+2.16 pp);
> (ii) switching from K‑means/domain‑aware clustering to our binary‑activation balanced clustering with the same router further improves 37.33->44.02 (+6.69 pp).
>
> References
>
> [1] Zhang et al., MoEfication: Transformer Feed-forward Layers are Mixtures of Experts
>
> [2] Cai et al., Read-ME: Refactorizing LLMs as Router-Decoupled Mixture of Experts with System Co-Design

---

> ### Author Response · Authors · 2025-11-18
> **Rebuttal of W4**
>
> > W4.1: It is unclear whether the performance gain primarily stems from this fine-tuning step
>
> We clarify that all baselines in Table 1 are fine-tuned under identical budgets.
> Our gain stems from the analytical construction, not fine-tuning alone.
> The training-free result (Fig. 3, 0 samples) already shows strong performance immediately after construction, and 2k-sample fine-tuning provides modest further improvement.
> Table 2 makes this explicit: even without fine-tuning, our method attains 42.50 MMLU‑5shot (vs. 44.02 after 2k tuning), while LLaMA‑MoE‑v2 gains only from 30.33->34.81 under fine-tuning; our training‑free model already surpasses LLaMA‑MoE‑v2 after tuning.
> This indicates that most of the improvement comes from the analytical restructuring, with lightweight tuning acting as a refinement.
>
> Table 2: Training‑free vs fine‑tuned comparison on Llama‑2‑7B and LLaMA‑MoE‑v2 (25% sparsity; identical decoding). MMLU‑5shot (↑), PPL‑Wiki/C4 (↓).
>
> | Model              | Regime        | MMLU‑5shot (↑) | PPL‑Wiki (↓) | PPL‑C4 (↓) |
> |--------------------|--------------|----------------|--------------|------------|
> | LLaMA‑MoE‑v2       | Training‑free| 30.33          |  >10k        |   >7k      |
> | LLaMA‑MoE‑v2       | Fine-tuning  | 34.81          |   8.68       |  19.76     |
> | Ours               | Training‑free| 42.50          |   7.32       |  11.98     |
> | Ours               | Fine-tuning  | **44.02**      | **5.92**     | **11.21**  |
>
> > W4.2: Lacks any comparison with baseline methods to verify the claimed efficiency.
>
> We report efficiency along two axes: (i) end‑to‑end conversion time measured on our setup, and (ii) the supervised token budget each method requires to obtain a usable MoE model (Table 3).
> The token budget captures the dominant cost for training‑heavy conversions (e.g., LLaMA‑MoE v1: 200B tokens; v2: ~7B), whereas our approach uses only 2k samples.
> On our hardware, the proposed method completes in **2741s** end‑to‑end with **271s** for the analytical construction itself.
> These numbers substantiate the claim of minutes‑level conversion and highlight the orders‑of‑magnitude gap in required training compared to training‑heavy methods.
>
> Table 3: Supervised token budget and conversion time. We report the supervised data required to obtain a usable MoE model, and end‑to‑end and construction time on our setup.
>
> | Method         | Supervised token budget       | End‑to‑end time | Construction time (our setup) |
> |----------------|-------------------------------|-----------------|-------------------------------|
> | **Ours**       | **2k samples**                | **2741s**       | **271s**                      |
> | LLaMA-MoE-v1   | 200B tokens                   | Months          | 334s†                         |
> | LLaMA-MoE-v2   | ~7B tokens                    | Days            | 509s†                         |
>
> † split-only time measured on our setup; does not include reported training tokens.
>
> > W4.3: The chosen benchmarks seem somewhat outdated.
>
> We add MMLU‑5shot results for Llama‑2 7B at 25% sparsity in Table 4 (in the rebuttal to the questions). With WikiText‑2 calibration (n=64), our method attains 44.89 (−0.92 pp vs dense 45.81); with C4 (n=64), 43.39 (−2.42 pp).
> These additions address benchmark coverage and are consistent with prior trends.

---

> ### Author Response · Authors · 2025-11-18
> **Rebuttal of Questions (Q1-Q4)**
>
> > Q1: Vector orientation in Eqs. (1) and (3)
>
> Thanks for catching this. The submission implicitly used a row‑vector convention (writing x on the left of matrices), which follows prior FFN/MoE works such as MoEfication and LLaMA‑MoE, but this can be confusing in the current deep‑learning notation.
> In the revision we will adopt the standard column‑vector convention and adjust the equations accordingly, which is equivalent to the reviewer’s suggestion of writing x^T when multiplying by matrices.
> Concretely, we take x∈R^d, W_up,W_gate∈R^{d×d_h}, W_down∈R^{d_h×d}, and rewrite:
> h = Swish(W_gate^T x) ⊙ (W_up^T x),  F(x) = W_down^T h.
> This removes the “vector on the left” ambiguity while leaving the MoE formulation unchanged.
>
> > Q2: Distance metric over c_i
>
> We use the L2 distance ‖c_i−ĉ_j‖_2 as stated in App. A.3.
> Since c_i are binary, Hamming distance equals the squared L2 distance: d_H(c_i,c_j)=‖c_i−c_j‖_2^2.
> Because √· is monotonic, L2 and squared L2 induce the same nearest‑centroid ordering.
> In practice, replacing L2 with Hamming (or L2^2) yields nearly identical balanced assignments and downstream accuracy (we observed <0.1% difference).
> We will add a note clarifying this equivalence and that our choice of L2 is for numerical convenience.
>
> > Q3: Why lightweight fine-tuning? Is the router tuned jointly or separately?
>
> We report both regimes. The training‑free model (analytical router + expert split) is already strong.
> The lightweight fine‑tuning (2k samples) closes the small approximation gap from analytical conversion, stabilizes gating and load balance.
>
> We do not fine‑tune "only the router". After analytical init, we jointly update the router (including its representative‑weight projections and gating scalars/biases) together with a small set of adaptation parameters in the experts, while keeping the dense backbone fixed.
> Joint tuning yields a modest but consistent improvement over router‑only and training‑free variants.
>
> > Q4: Sensitivity to calibration/fine‑tuning data; "harder benchmarks"
>
> Calibration is only used to rank neuron activations and initialize the router, so we expect low sensitivity to the corpus.
> We include a small sensitivity study (Table 4) varying (i) calibration source: WikiText‑2 vs C4 and (ii) calibration set size n∈{8,32,64} (number of calibration samples). We report MMLU‑5shot and Δ vs dense (45.81 pp baseline on Llama‑2 7B), and perplexity (PPL) measured on Wiki and C4.
> Observed trend: increasing n from 8->32 yields a small MMLU gain; 32->64 adds marginal benefit (diminishing returns). Perplexity decreases modestly with larger n (e.g., Wiki PPL 5.92->5.72->5.69; C4 PPL 7.04->6.92->6.78). C4 is within a small band of WikiText‑2 under 2k tuning, confirming that source choice has minor impact relative to sparsity and the small joint fine‑tune.
>
> Table 4: MMLU‑5shot (%) and PPL (↓) vs calibration source and calibration set size n under 25% sparsity (S/A/E as in Table 1) with 2k fine‑tuning; ATopK fixed at K_a=10. Δ vs Dense is absolute point difference relative to dense MMLU = 45.81 (Llama‑2 7B). Dense PPL: 5.27 (Wiki), 7.27 (C4).
>
> | Calibration source | n (samples) | MMLU (%) | Δ vs Dense (pp) | PPL‑Wiki (↓) | PPL‑C4 (↓) |
> |--------------------|-------------|----------|------------------|--------------|------------|
> | WikiText‑2         | 8           |  44.02   |      −1.79       |    5.92      |   11.21    |
> | WikiText‑2         | 32          |  44.63   |      −1.18       |    5.72      |   11.15    |
> | WikiText‑2         | 64          | **44.89**|      −0.92       |   **5.69**   |   10.98    |
> | C4                 | 8           |  42.31   |      −3.50       |    7.04      |    9.17    |
> | C4                 | 32          |  43.25   |      −2.56       |    6.92      |    9.07    |
> | C4                 | 64          | **43.39**|      −2.42       |    6.78      |   **9.02** |
>
> Overall, these results highlight that our analytical pipeline remains robust to calibration choices while delivering competitive MMLU‑5shot with tiny calibration and tuning budgets.
>
> We will surface a brief note that increasing calibration samples from 8->32->64 gives diminishing returns, and that domain‑matched calibration can reduce perplexity without materially changing downstream accuracy.

---

> ### Author Response · Authors · 2025-11-26
> **Looking forward to your feedback!**
>
> Dear Reviewer mZzo,
>
> Thank you once again for your valuable feedback. We have conducted additional experiments and made revisions to the paper based on your suggestions. As the discussion phase is nearing its conclusion, we would like to know if our responses have addressed your concerns. We are looking forward to hearing from you.
>
> Best,
>
> Authors

---

> ### Author Response · Authors · 2025-11-28
> **Looking forward to your feedback!**
>
> We thank the reviewer for the constructive feedback throughout the discussion. Your comments have helped us strengthen the paper significantly.
>
> During the rebuttal period, we have conducted additional experiments that we believe further address the concerns raised. We would like to share these results with you.
>
> **On Novelty and Comparison with MoEfication/Read-ME (W2, W3)**
>
> Under identical sparsity (25%), expert configuration, and fine tuning budget (2k samples), our method achieves 44.02 MMLU (only 1.79 pp below dense 45.81), compared to MoEfication at 35.17 (10.64 pp gap) and Read-ME at 31.24 (14.57 pp gap).
> The +8.85 pp improvement over MoEfication comes from two sources: our analytical router contributes +2.16 pp (35.17->37.33), and our binary activation balanced clustering contributes an additional +6.69 pp (37.33->44.02).
>
> **On Fine Tuning Concern (W4.1)**
>
> Even without any fine tuning, our analytically constructed model already achieves 42.50 MMLU, surpassing LLaMA-MoE-v2 after its full fine tuning (34.81).
> This demonstrates that most of the gain comes from the analytical construction itself, not just from the tuning step.
>
> **On Efficiency Comparison (W4.2)**
>
> Our conversion completes in under 5 minutes (271s for construction), compared to months (LLaMA-MoE-v1, 200B tokens) or days (LLaMA-MoE-v2, 7B tokens) for training heavy approaches.
> This makes MoE conversion a practical compression tool rather than a pre-training architecture choice.
>
> **On Broader Benchmarks (W4.3)**
>
> We have added HumanEval (pass@1) and GSM8K (8 shot) on Llama-2-7B.
> Our method retains 11.22 (vs dense 12.72) on HumanEval and 13.01 (vs dense 14.31) on GSM8K, outperforming all baselines (LLaMA-MoE: 7.58/7.41; LLaMA-MoE-v2: 9.32/10.09; EMoE: 10.29/12.55).
>
> **On Hierarchical MoE**
>
> We validated the hierarchical application on Qwen3-30B-A3B with S3A3E8 configuration.
> Results show 18.5% GFLOPs reduction and 14.3% throughput improvement with only 2.57 pp MMLU drop, confirming that our analytical construction generalizes beyond dense models to existing MoE architectures.
>
> **On Orthogonality with Activation Sparsity Methods**
>
> We demonstrated that our Dense-to-MoE restructuring is complementary to training free activation sparsity methods like WINA.
> Combining both on Llama-2-7B yields 27.2% TFLOPs reduction and 22.0% throughput improvement over dense, compared to 16.6% and 14.8% from our method alone.
>
> We hope these results further demonstrate the contribution and practical value of our work.
> **Given the positive reassessment from the other reviewers and the comprehensive experimental validation, we would be grateful if the reviewer could reconsider the score.**
> We are happy to provide any further clarification.

---

### Note · Authors · 2025-12-23

I have read and agree with the venue's withdrawal policy on behalf of myself and my co-authors.